# REFLECTED SCHRÖDINGER BRIDGE FOR CONSTRAINED GENERATIVE MODELING

## ABSTRACT

Diffusion models have become the go-to method for large-scale generative models in real-world applications. These applications often involve data distributions confined within bounded domains, typically requiring ad-hoc thresholding techniques for boundary enforcement. Reflected diffusion models (Lou & Ermon, 2023) aim to enhance generalizability by generating the data distribution through a backward process governed by reflected Brownian motion. However, reflected diffusion models may not easily adapt to diverse domains without the derivation of proper diffeomorphic mappings and do not guarantee optimal transport properties. To overcome these limitations, we introduce the Reflected Schrödinger Bridge algorithm—an entropy-regularized optimal transport approach tailored for generating data within diverse bounded domains. We derive elegant reflected forward-backward stochastic differential equations with Neumann and Robin boundary conditions, extend divergence-based likelihood training to bounded domains, and explore natural connections to entropic optimal transport for the study of approximate linear convergence—a valuable insight for practical training. Our algorithm yields robust generative modeling in diverse domains, and its scalability is demonstrated in real-world constrained generative modeling through standard image benchmarks.

## 1 INTRODUCTION

Iterative refinement is key to the unprecedented success of diffusion models. They exhibit statistical efficiency (Koehler et al., 2023) and reduced dimensionality dependence (Vono et al., 2022), driving innovation in image, audio, video, and molecule synthesis (Dhariwal & Nichol, 2022; Ho et al., 2022; Hoogeboom et al., 2022; Bunne et al., 2023). However, diffusion models do not inherently guarantee optimal transport properties (Lavenant & Santambrogio, 2022) and often result in slow inference (Ho et al., 2020; Salimans & Ho, 2022; Lu et al., 2022). Furthermore, the consistent reliance on Gaussian priors imposes limitations on the application potential and sacrifices the efficiency when the data distribution significantly deviates from the Gaussian prior.

The predominant method for fast inference originates from the field of optimal transport (OT). Notably, the (static) iterative proportional fitting (IPF) algorithm (Kullback, 1968; Ruschendorf, 1995) addresses this challenge by employing alternating projections onto each marginal distribution. This algorithm has showcased impressive performance in low-dimensional contexts (Chen & Georgiou, 2016; Pavon et al., 2021; Caluya & Halder, 2022). In contrast, the Schrödinger bridge (SB) problem (Léonard, 2014) introduces a principled framework for the dynamic treatment of entropy-regularized optimal transport (EOT) (Villani, 2003; Peyré & Cuturi, 2019). Recent advances (De Bortoli et al., 2021; Chen et al., 2022b) have pushed the frontier of IPFs to (ultra-)high-dimensional generative models using deep neural networks (DNNs) and have generated straighter trajectories; Additionally, SBs based on Gaussian process (Vargas et al., 2021) demonstrates great promise in robustness and scalability; Bridge matching methods (Shi et al., 2023; Peluchetti, 2023) also offers promising alternatives for solving complex dynamic SB problems.

Real-world data, such as pixel values in images, often exhibits bounded support. To address this challenge, a common practice involves the use of thresholding techniques (Ho et al., 2020) to guide the sampling process towards the intended domain of simple structures. Lou & Ermon (2023) introduced reflected diffusion models that employ reflected Brownian motion on constrained domains such as hypercubes and simplex. However, constrained domains on general Euclidean space with

optimal transport guarantee are still not well developed. Moreover, Lou & Ermon (2023) relies on a uniform prior based on variance-exploding (VE) SDE to derive closed-form scores, and the popular variance-preserving (VP) SDE is not fully exploited.

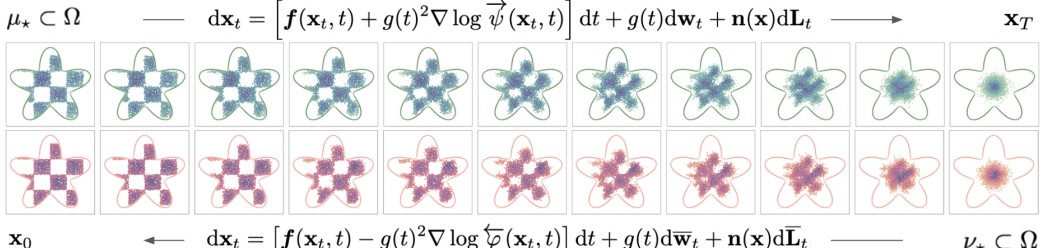

$$\mu_\star \subset \Omega \qquad ---\qquad d\mathbf{x}_t = \left[\boldsymbol{f}(\mathbf{x}_t, t) + g(t)^2 \nabla \log \overrightarrow{\psi}(\mathbf{x}_t, t)\right] dt + g(t)d\mathbf{w}_t + \mathbf{n}(\mathbf{x})d\mathbf{L}_t \qquad \longrightarrow \qquad \mathbf{x}_T$$

$$\mathbf{x}_0 \qquad \longleftarrow \qquad d\mathbf{x}_t = \left[\boldsymbol{f}(\mathbf{x}_t, t) - g(t)^2 \nabla \log \overleftarrow{\varphi}(\mathbf{x}_t, t)\right] dt + g(t)d\overline{\mathbf{w}}_t + \mathbf{n}(\mathbf{x})d\overline{\mathbf{L}}_t \qquad --- \qquad \nu_\star \subset \Omega$$

Figure 1: Constrained generative modeling via reflected forward-backward SDEs.

To bridge this gap, we propose the *Reflected Schrödinger Bridge* (SB) to model the transport between any smooth distributions with bounded support. We derive novel reflected forward-backward stochastic differential equations (reflected FB-SDEs) with Neumann and Robin boundary conditions and extend the divergence-based likelihood training to ensure its confinement within any smooth boundaries. We further establish connections between reflected FB-SDEs and EOT on bounded domains, where the latter facilitates the theoretical understanding by analyzing the convergence of the dual, potentials, and couplings on bounded domains. Notably, our analysis provides the first non-geometric approach to study the uniform-in-time stability w.r.t. the marginals and is noteworthy in its own right. We empirically validate our algorithm on 2D examples and standard image benchmarks, showcasing its promising performance in generative modeling over constrained domains. The flexible choices on the priors allow us to choose freely between VP-SDE and VE-SDE.

Regarding *related works* on constrained sampling and generation, we refer readers to Appendix A.

## 2 PRELIMINARIES

Diffusion models (Song et al., 2021b) have achieved tremendous progress in real-world applications, such as image and text-to-image generation. However, real-world data (such as the bounded pixel space in images) often comes with bounded support. As an illustration within the computer vision field, practitioners often employ ad-hoc thresholding techniques to project the data to the desired space, which inevitably affects the theoretical understanding and hinders future updates.

To generalize these techniques in a principled framework, Lou & Ermon (2023) utilized reflected Brownian motion to train explicit score-matching loss in bounded domains. They first perturb the data with a sequence of noise and then propose to generate the constrained data distribution through the corresponding reflected backward process (Williams, 1987; Cattiaux, 1988).

$$d\mathbf{x}_t = \boldsymbol{f}(\mathbf{x}_t, t)dt + g(t)d\mathbf{w}_t + d\mathbf{L}_t, \qquad\qquad \mathbf{x}_0 \sim p_{\text{data}} \subset \Omega \tag{1a}$$

$$d\mathbf{x}_t = \left[\boldsymbol{f}(\mathbf{x}_t, t) - g(t)^2 \nabla \log p_t(\mathbf{x}_t)\right] dt + g(t)d\overline{\mathbf{w}}_t + d\overline{\mathbf{L}}_t, \ \mathbf{x}_T \sim p_{\text{prior}} \subset \Omega \tag{1b}$$

where $\Omega$ is the state space in $\mathbb{R}^d$; $\boldsymbol{f}(\mathbf{x}_t, t)$ and $g(t)$ are the vector field and the diffusion term, respectively; $\mathbf{w}_t$ is the Brownian motion; $\overline{\mathbf{w}}_t$ is another independent Brownian motion from time $T$ to 0; $\mathbf{L}_t$ and $\overline{\mathbf{L}}_t$ are the local time to confine the particle within the domain and are defined in Eq.(18); the marginal density at time $t$ for the forward process (1a) is denoted by $p_t$. $\nabla \log p_t(\cdot)$ is the score function at time $t$, which is often approximated by a neural network model $s_\theta(\cdot, t)$. Given proper score approximations, the data distribution $p_{\text{data}}$ can be generated from the backward process (1b).

## 3 REFLECTED SCHRÖDINGER BRIDGE

Although reflected diffusion models have demonstrated empirical success in image applications on hypercubes, extensions to general domains with optimal-transport guarantee remain limited (Lavenant & Santambrogio, 2022). Notably, the forward process (1a) requires a long time $T$ to approach the prior distribution, which inevitably leads to a slow inference (De Bortoli et al., 2021). To solve that problem, the dynamic SB problem on a bounded domain $\Omega$ proposes to solve

$$\inf_{\mathbb{P} \in \mathcal{D}(\mu_\star, \nu_\star)} \text{KL}(\mathbb{P} \| \mathbb{Q}), \tag{2}$$

where the coupling $\mathbb{P}$ belongs to the path space $\mathcal{D}(\mu_\star, \nu_\star) \subset C(\Omega, [0, T])$ with marginal measures $\mu_\star$ at time $t = 0$ and $\nu_\star$ at $t = T$; $\mathbb{Q}$ is the prior path measure, such as the measure induced by the path of the reflected Brownian motion or Ornstein-Uhlenbeck (OU) process. From the perspective of stochastic control, the dynamical SBP aims to minimize the cost along the reflected process

$$\inf_{\boldsymbol{u} \in \mathcal{U}} \mathbb{E} \left\{ \int_0^T \frac{1}{2} \|\boldsymbol{u}(\mathbf{x}_t, t)\|_2^2 \mathrm{d}t \right\}$$

$$\text{s.t. } \mathrm{d}\mathbf{x}_t = [\boldsymbol{f}(\mathbf{x}_t, t) + g(t)\boldsymbol{u}(\mathbf{x}_t, t)] \, \mathrm{d}t + \sqrt{2\varepsilon} g(t) \mathrm{d}\mathbf{w}_t + \mathbf{n}(\mathbf{x}_t) \mathrm{d}\mathbf{L}_t, \quad (3)$$

$$\mathbf{x}_0 \sim \mu_\star, \ \mathbf{x}_T \sim \nu_\star, \ \mathbf{x}_t \in \Omega, \quad \text{for any } t \in [0, T]$$

where $\mathcal{U}$ is a set of control functions; $\varepsilon$ is the entropic regularizer for EOT; $\mathbf{n}(\mathbf{x})$ is an inner unit normal vector at $\mathbf{x} \in \partial\Omega$ and $\mathbf{0}$ for $\mathbf{x} \in \Omega$; the expectation follows from the density $\rho(\mathbf{x}, t)$. Simulation demos of the reflected SDEs are shown in Figure 2.

To derive the reflected FB-SDEs and training scheme, we first present standard assumptions on the regularity properties (Øksendal, 2003), as well as the smoothness of measure (Chen et al., 2022a;b) and boundary (Lamperski, 2021):

**Assumption A1** (Regularity on drift and diffusion). *The drift $\boldsymbol{f}$ and diffusion term $g > 0$ satisfy the Lipschitz and linear growth condition.*

**Assumption A2** (Smooth boundary). *The domain $\Omega$ is bounded and has a smooth boundary.*

Extensions to general convex domains (with corners) are also studied in Lamperski (2021).

**Assumption A3** (Smooth measure). *The probability measures $\mu_\star$ and $\nu_\star$ are smooth in the sense that the energy functions $U_\star = -\nabla \log \frac{\mathrm{d}\mu_\star}{\mathrm{d}\mathbf{x}}$ and $V_\star = -\nabla \log \frac{\mathrm{d}\nu_\star}{\mathrm{d}\mathbf{x}}$ are differentiable.*

### 3.1 REFLECTED FORWARD-BACKWARD STOCHASTIC DIFFERENTIAL EQUATIONS

Following the tradition in mechanics (Pavliotis, 2014), we rewrite the reflected SBP as follows

$$\inf_{\boldsymbol{u} \in \mathcal{U}} \int_0^T \int_\Omega \frac{1}{2} \rho \|\boldsymbol{u}\|_2^2 \mathrm{d}\mathbf{x} \mathrm{d}t$$

$$\text{s.t. } \frac{\partial \rho}{\partial t} + \nabla \cdot \mathbf{J}|_{\mathbf{x} \in \Omega} = 0, \ \langle \mathbf{J}, \mathbf{n} \rangle|_{\mathbf{x} \in \partial\Omega} = 0, \quad (4)$$

where $\mathbf{J}$ is the probability flux of continuity equation $\mathbf{J} \equiv \rho(\boldsymbol{f} + g\boldsymbol{u}) - \varepsilon g^2 \nabla \rho$ (Pavliotis, 2014).

We next solve the objectives with a Lagrangian multiplier: $\phi(\mathbf{x}, t)$. Applying the Stokes theorem with details presented in appendix B.1, we have

$$\mathcal{L}(\rho, \boldsymbol{u}, \phi) = \underbrace{\int_0^T \int_\Omega \left( \frac{1}{2} \rho \|\boldsymbol{u}\|_2^2 - \rho \frac{\partial \phi}{\partial t} - \langle \nabla \phi, \mathbf{J} \rangle \right) \mathrm{d}\mathbf{x} \mathrm{d}t}_{\overline{\mathcal{L}}(\rho, \boldsymbol{u}, \phi)} + \underbrace{\int_\Omega \phi \rho|_{t=0}^T \mathrm{d}\mathbf{x}}_{\text{constant term w.r.t. } \boldsymbol{u}} + \underbrace{\int_0^T \int_{\partial\Omega} \langle \mathbf{J}, \mathbf{n} \rangle \mathrm{d}\sigma(\mathbf{x}) \mathrm{d}t}_{:=0 \text{ by Eq.}(4)}.$$

Minimizing $\mathcal{L}$ with respect to $\boldsymbol{u}$, we can obtain $\boldsymbol{u}^\star = g\nabla\phi$. Further applying the Cole-Hopf transform $\overrightarrow{\psi}(\mathbf{x}, t) = \exp\left(\frac{\phi(\mathbf{x}, t)}{2\varepsilon}\right)$ and setting $\overline{\mathcal{L}}(\rho, \boldsymbol{u}^\star, \phi) = 0$, we derive the *backward Kolmogorov equation* with *Neumann boundary* conditions

$$\begin{cases} \frac{\partial \overrightarrow{\psi}}{\partial t} + \varepsilon g^2 \Delta \overrightarrow{\psi} + \langle \nabla \overrightarrow{\psi}, \boldsymbol{f} \rangle = 0 & \text{in } \Omega \\ \langle \nabla \overrightarrow{\psi}, \mathbf{n} \rangle = 0 & \text{on } \partial\Omega. \end{cases}$$

Next we define $\overleftarrow{\varphi} = \rho^\star / \overrightarrow{\psi}$, where $\rho^\star$ is the optimal density of Eq.(3) given $\boldsymbol{u}^\star$. We arrive at the *forward Kolmogorov equation* with the *Robin boundary* condition

$$\begin{cases} \partial_t \overleftarrow{\varphi} + \nabla \cdot \left( \overleftarrow{\varphi} \boldsymbol{f} - \varepsilon g^2 \nabla \overleftarrow{\varphi} \right) = 0 & \text{in } \Omega \\ \langle \overleftarrow{\varphi} \boldsymbol{f} - \varepsilon g^2 \nabla \overleftarrow{\varphi}, \mathbf{n} \rangle = 0 & \text{on } \partial\Omega. \end{cases}$$

Despite the elegance, solving PDEs in high dimensions often poses significant challenges due to the curse of dimensionality (Han et al., 2019). To overcome these challenges, we resort to presenting a set of reflected FB-SDEs:

**Theorem 1.** *Consider a* Schrödinger *(PDE) system* *with Neumann and Robin boundary conditions*

$$\begin{cases} \frac{\partial \overrightarrow{\psi}}{\partial t} + \langle \nabla \overrightarrow{\psi}, \boldsymbol{f} \rangle + \varepsilon g^2 \Delta \overrightarrow{\psi} = 0 \\ \frac{\partial \overleftarrow{\varphi}}{\partial t} + \nabla \cdot (\overleftarrow{\varphi} \boldsymbol{f}) - \varepsilon g^2 \Delta \overleftarrow{\varphi} = 0 \end{cases} \quad s.t. \ \ \langle \nabla \overrightarrow{\psi}, \mathbf{n} \rangle|_{\mathbf{x} \in \partial \Omega} = 0, \langle \boldsymbol{f} \overleftarrow{\varphi} - \varepsilon g^2 \nabla \overleftarrow{\varphi}, \mathbf{n} \rangle|_{\mathbf{x} \in \partial \Omega} = 0. \quad (5)$$

*Solving the PDE system gives rise to the reflected FB-SDEs with* $\mathbf{x}_t \in \Omega$

$$\mathrm{d}\mathbf{x}_t = \left[ \boldsymbol{f}(\mathbf{x}_t, t) + 2\varepsilon g(t)^2 \nabla \log \overrightarrow{\psi}(\mathbf{x}_t, t) \right] \mathrm{d}t + \sqrt{2\varepsilon}g(t)\mathrm{d}\mathbf{w}_t + \mathbf{n}(\mathbf{x})\mathrm{d}\mathbf{L}_t, \ \ \mathbf{x}_0 \sim \mu_\star, \quad (6a)$$

$$\mathrm{d}\mathbf{x}_t = \left[ \boldsymbol{f}(\mathbf{x}_t, t) - 2\varepsilon g(t)^2 \nabla \log \overleftarrow{\varphi}(\mathbf{x}_t, t) \right] \mathrm{d}t + \sqrt{2\varepsilon}g(t)\mathrm{d}\overline{\mathbf{w}}_t + \mathbf{n}(\mathbf{x})\mathrm{d}\overline{\mathbf{L}}_t, \ \ \mathbf{x}_T \sim \nu_\star. \quad (6b)$$

*The connection to the probability flow ODE is also studied and presented in section B.2.*

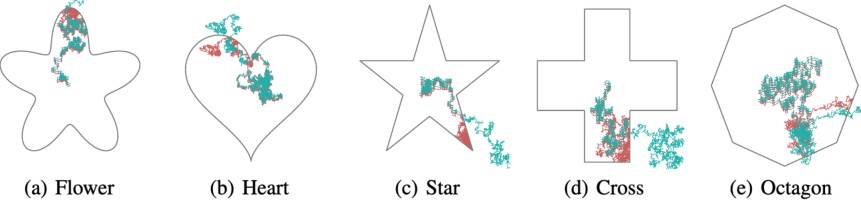

|     |     |     |     |     |
| --- | --- | --- | --- | --- |
| (a) Flower | (b) Heart | (c) Star | (d) Cross | (e) Octagon |

Figure 2: Reflected OU processes (**reflected** v.s. unconstrained), driven by the same Brownian motion, excluding the reflections. All boundary curves have properly defined unit vectors.

### 3.2 LIKELIHOOD TRAINING

It is worth mentioning that the reflected FB-SDE (6) is not directly accessible due to the unknown control variables $(\nabla \log \overrightarrow{\psi}, \nabla \log \overleftarrow{\varphi})$. To tackle this issue, a standard tool is the (nonlinear) Feynman-Kac formula (Ma & Yong, 2007; Karatzas & Shreve, 1998), which leads to a stochastic representation.

**Proposition 1** (Feynman-Kac representation). *Assume assumptions A1-A2 hold.* $\overleftarrow{\varphi}$ *satisfies a PDE (5) and* $\mathbf{x}_t$ *follows from a diffusion (6a). Define* $\overrightarrow{y}_t \equiv \overrightarrow{y}(\mathbf{x}_t, t) = \log \overrightarrow{\psi}(\mathbf{x}_t, t)$ *and* $\overleftarrow{y}_t \equiv \overleftarrow{y}(\mathbf{x}_t, t) = \log \overleftarrow{\varphi}(\mathbf{x}_t, t)$. *Then* $\overleftarrow{y}_s$ *admits a stochastic representation*

$$\overleftarrow{y}_s = \mathbb{E}\left[ \overleftarrow{y}_T - \int_s^T \underbrace{\left( \frac{1}{2}\|\overleftarrow{\mathbf{z}}_t\|_2^2 + \nabla \cdot (\overleftarrow{g}\,\mathbf{z}_t - \boldsymbol{f}) + \langle \overleftarrow{\mathbf{z}}_t, \overrightarrow{\mathbf{z}}_t \rangle \right)}_{\zeta(\mathbf{x}_t, t)}\mathrm{d}t - \mathrm{d}\overleftarrow{\mathbf{L}}_t \,\Big|\, \mathbf{x}_s = \boldsymbol{x}_s \right],$$

*on* $[0, T] \times \Omega$; $\overrightarrow{\mathbf{z}}_t \equiv \overrightarrow{\mathbf{z}}(\mathbf{x}_t, t) = g\nabla\overrightarrow{y}_t$, $\overleftarrow{\mathbf{z}}_t \equiv \overleftarrow{\mathbf{z}}(\mathbf{x}_t, t) = g\nabla\overleftarrow{y}_t$, $\mathrm{d}\overleftarrow{\mathbf{L}}_t = \frac{1}{g}\langle \overleftarrow{\mathbf{z}}_t, \mathbf{n}_t \rangle \mathrm{d}\mathbf{L}_t$.

**Sketch of proof** The proof primarily relies on Theorem 3 from Chen et al. (2022b) and applies (generalized) Itô's lemma to $\overleftarrow{y}_t$ using (5) and (6a). The difference is to incorporate the generalized Itô's lemma (Bubeck et al., 2018; Lamperski, 2021) to address the local time of $\mathbf{x}_t$ at the boundary $\partial\Omega$. Subsequently, our analysis establishes that $\overleftarrow{y}_s - \int_{s_1}^s \zeta(\mathbf{x}_t, t)$, where $s \in [s_1, T]$, is a martingale within the domain $\Omega$, which concludes our proposition. $\qquad\square$

A direct application of the proposition is to obtain the log-likelihood $\overleftarrow{y}_0$ given data points $\mathbf{x}_0$. With parametrized models $(\overrightarrow{\mathbf{z}}_t^\theta, \overleftarrow{\mathbf{z}}_t^\omega)$ to approximate $(\overrightarrow{\mathbf{z}}_t, \overleftarrow{\mathbf{z}}_t)$, we can optimize the backward score function $\overleftarrow{\mathbf{z}}_t^\omega$ through the forward loss function $\mathcal{L}(\mathbf{x}_0; \omega)$ in Algorithm 1. Regarding the forward-score estimation, similar to Theorem 11 (Chen et al., 2022b), the symmetric property of the reflected SB also enables to optimize $\overrightarrow{\mathbf{z}}_t$ via the backward loss function $\mathcal{L}(\mathbf{x}_T; \theta)$.

By the data processing inequality, our loss function provides a lower bound of the log-likelihood, which resembles the evidence lower bound (ELBO) in variational inference (Song et al., 2021a). We can expect a smaller variational gap given more accurate parametrized models.

When the domain is taken to be $\Omega = \mathbb{R}^d$, the aforementioned solvers become equivalent to the loss function (18-19) presented in Chen et al. (2022b).

### 3.3 CONNECTIONS TO THE IPF ALGORITHM

Similar in spirit to Theorem 3 of Song et al. (2021a), Algorithm 1 results in an elegant half-bridge solver ($\mu_\star \to \nu_\star$ v.s. $\mu_\star \leftarrow \nu_\star$) to approximate the primal formulation (Nutz, 2022) of the dynamic

---

**Algorithm 1** One iteration of the backward-forward score function solver to optimize $(\overrightarrow{\mathbf{z}}_t^\theta, \overleftarrow{\mathbf{z}}_t^\omega)$ with the reflection implemented in Algorithm 4. We cache the trajectories following De Bortoli et al. (2021) to avoid expensive computational graphs. In practice, $\mathbb{E}[\log \overleftarrow{y}_T]$ and $\mathbb{E}[\log \overrightarrow{y}_0]$ are often omitted to facilitate training (Chen et al., 2022b).

$$\mathcal{L}(\mathbf{x}_0; \omega) = \mathbb{E}[\log \overleftarrow{y}_T] - \int_0^T \mathbb{E}_{\mathbf{x}_t \sim (6a)}\left[\left(\frac{1}{2}\|\overleftarrow{\mathbf{z}}_t^\omega\|_2^2 + g\nabla \cdot \overleftarrow{\mathbf{z}}_t^\omega + \langle \overrightarrow{\mathbf{z}}_t^\theta, \overleftarrow{\mathbf{z}}_t^\omega \rangle\right)\mathrm{d}t + \mathrm{d}\overleftarrow{\mathbf{L}}_t^\omega \Big| \mathbf{x}_0 = \mathbf{x}_0\right]$$

$$\mathcal{L}(\mathbf{x}_T; \theta) = \mathbb{E}[\log \overrightarrow{y}_0] - \int_0^T \mathbb{E}_{\mathbf{x}_t \sim (6b)}\left[\left(\frac{1}{2}\|\overrightarrow{\mathbf{z}}_t^\theta\|_2^2 + g\nabla \cdot \overrightarrow{\mathbf{z}}_t^\theta + \langle \overleftarrow{\mathbf{z}}_t^\omega, \overrightarrow{\mathbf{z}}_t^\theta \rangle\right)\mathrm{d}t + \mathrm{d}\overrightarrow{\mathbf{L}}_t^\theta \Big| \mathbf{x}_T = \mathbf{x}_T\right],$$

where $\mathrm{d}\overleftarrow{\mathbf{L}}_t^\omega = \frac{1}{g}\langle \overleftarrow{\mathbf{z}}_t^\omega, \mathbf{n}_t \rangle \mathrm{d}\mathbf{L}_t$ and $\mathrm{d}\overrightarrow{\mathbf{L}}_t^\theta = \frac{1}{g}\langle \overrightarrow{\mathbf{z}}_t^\theta, \mathbf{n}_t \rangle \mathrm{d}\overline{\mathbf{L}}_t$. (6a) (respectively, (6b)) is approximated via $\overrightarrow{\mathbf{z}}_t^\theta$ (respectively, $\overleftarrow{\mathbf{z}}_t^\omega$).

---

Schrödinger bridge (2) (De Bortoli et al., 2021; Vargas et al., 2021):

**Dynamic Primal IPF** $\quad \mathbb{P}_{2k} = \underset{\mathbb{P} \in \mathcal{D}(\cdot, \nu_\star)}{\arg\min} \mathrm{KL}(\mathbb{P}\|\mathbb{P}_{2k-1}), \quad \mathbb{P}_{2k+1} = \underset{\mathbb{P} \in \mathcal{D}(\mu_\star, \cdot)}{\arg\min} \mathrm{KL}(\mathbb{P}\|\mathbb{P}_{2k}), \quad$ (7)

which is also known as the dynamic IPF algorithm (also known as Sinkhorn algorithm) (Ruschendorf, 1995; De Bortoli et al., 2021). Consider the disintegration of the path measure $\mathbb{P} = \pi \otimes \mathbb{P}^{\mu_\star, \nu_\star}$

$$\mathbb{P}(\cdot) = \iint_{\Omega^2} \mathbb{P}^{\mathbf{x}_0, \mathbf{x}_T}(\cdot)\pi(\mathrm{d}\mathbf{x}_0, \mathrm{d}\mathbf{x}_T), \tag{8}$$

where $\mathbb{P}^{\mathbf{x}_0, \mathbf{x}_T} \in \mathbb{P}^{\mu_\star, \nu_\star}$ is a diffusion bridge from $\mathbf{x}_0 = \mathbf{x}_0$ to $\mathbf{x}_T = \mathbf{x}_T$, $\pi \in \Pi(\mu_\star, \nu_\star)$ and the product space $\Pi(\mu_\star, \nu_\star) \subset \Omega^2$ denotes the space of couplings with the first and second marginals following from $\mu_\star$ and $\nu_\star$, respectively. Now project the path space $\mathcal{D}$ to the product space $\Pi$. We have the static IPF algorithm in the primal formulation:

**Static Primal IPF** $\quad \pi_{2k} = \underset{\pi \in \Pi(\cdot, \nu_\star)}{\arg\min} \mathrm{KL}(\pi\|\pi_{2k-1}), \quad \pi_{2k+1} = \underset{\pi \in \Pi(\mu_\star, \cdot)}{\arg\min} \mathrm{KL}(\pi\|\pi_{2k}). \quad$ (9)

## 4 CONVERGENCE ANALYSIS VIA ENTROPIC OPTIMAL TRANSPORT

The dynamic IPF algorithm offers an efficient training scheme to fit marginals in high-dimensional problems. However, the understanding of the convergence remains unclear to the machine learning community. To get around this issue, we leverage the progress from the static optimal transport on bounded domains and costs (Carlier, 2022; Chen et al., 2016; Deligiannidis et al., 2021).

Our analysis is illustrated as follows: We first draw connections between dynamic and static (primal) IPFs by projecting the path space $\mathcal{D}$ to the product space $\Pi$ and then show the equivalence between the dual and primal formulations. Next, we perturb the marginals (in terms of energy functions) and show the approximate linear convergence of the dual, potential, and then static couplings. The convergence of dynamic couplings can be expected given a reasonable estimate of diffusion bridge.

Dynamic Primal IPF (7) $\xleftrightarrow[\text{Projection}]{\text{Disintegration}}$ Static Primal IPF (9) $\xleftrightarrow[\text{Lemma 1}]{\text{Equivalence}(C.5)}$ Static Dual IPF (13)

### 4.1 EQUIVALENCE BETWEEN DYNAMIC SBP AND STATIC SBP

Assuming the solutions exist, the disintegration of measures implies that the equivalence of solutions between the dynamic and static SBPs (Léonard, 2014):

**Dynamic SBP** $\quad \mathbb{P}_\star = \underset{\mathbb{P} \in \mathcal{D}(\mu_\star, \nu_\star)}{\arg\min} \mathrm{KL}(\mathbb{P}\|\mathbb{Q}) \Longleftrightarrow \pi_\star = \underset{\pi \in \Pi(\mu_\star, \nu_\star)}{\arg\min} \mathrm{KL}(\pi\|\mathcal{G}), \quad$ **Static**

where $\pi$ (respectively, $\mathcal{G}$) is the projection of the path measure $\mathbb{P}$ (respectively, $\mathbb{Q}$) on the product space at $t = 0$ and $T$; $\mathrm{d}\mathcal{G} \propto e^{-c_\varepsilon}\mathrm{d}(\mu_\star \otimes \nu_\star)$; $c_\varepsilon$ is a cost function. Both the dynamic and static SBP formulations yield structure properties (see the Born's formula in Léonard (2014)) and enables to represent Schrödinger bridges $\mathbb{P}_\star$ and $\pi_\star$ using Schrödinger potentials $\varphi_\star$ and $\psi_\star$:

**Dynamic Struture** $\quad \mathrm{d}\mathbb{P}_\star = e^{\varphi_\star(\mathbf{x}) + \psi_\star(\mathbf{y})}\mathrm{d}\mathbb{Q} \Longleftrightarrow \mathrm{d}\pi_\star(\mathbf{x}, \mathbf{y}) = e^{\varphi_\star(\mathbf{x}) + \psi_\star(\mathbf{y})}\mathrm{d}\mathcal{G}. \quad$ **Static** (10)

Moreover, the summation $\varphi_\star \oplus \psi_\star$ is unique such that $(\varphi_\star + a) \oplus (\psi_\star - a)$ is also viable for any $a$.

This static structural representation establishes a connection between the static SBP and entropic optimal transport (EOT) with a unit entropy regularizer (Chen et al., 2023), and the latter results in an efficient scheme to compute the optimal coupling:

$$\inf_{\pi \in \Pi(\mu_\star, \nu_\star)} \iint_{\Omega^2} c_\varepsilon(\mathbf{x}, \mathbf{y}) \pi(\mathrm{d}\mathbf{x}, \mathrm{d}\mathbf{y}) + \mathrm{KL}(\pi \| \mu_\star \otimes \nu_\star).$$

### 4.2 DUALITY FOR SCHRÖDINGER BRIDGES AND APPROXIMATIONS

The Schrödinger bridge is a constrained optimization problem and possesses a computation-friendly dual formulation. Moreover, the duality gap is zero under probability measures (Léonard, 2001).

**Lemma 1** (Duality (Nutz, 2022)). *Given assumptions A1-A3, the dual via potentials $(\varphi, \psi)$ follows*

$$\min_{\pi \in \Pi(\mu_\star, \nu_\star)} \mathrm{KL}(\pi | \mathcal{G}) = \max_{\varphi, \psi} G(\varphi, \psi), \quad G(\varphi, \psi) := \mu_\star(\varphi) + \nu_\star(\psi) - \iint_{\Omega^2} e^{\varphi \oplus \psi} \mathrm{d}\mathcal{G} + 1, \quad (11)$$

*where $\mu_\star(\varphi) = \int_\Omega \varphi \mathrm{d}\mu_\star$, $\nu_\star(\psi) = \int_\Omega \psi \mathrm{d}\nu_\star$, $\varphi \in L^1(\mu_\star)$, and $\psi \in L^1(\nu_\star)$.*

An effective solver is to maximize the dual $G$ via $\varphi_{k+1} = \arg\max_{\varphi \in L^1(\mu_\star)} G(\varphi, \psi_k)$ and $\psi_{k+1} = \arg\max_{\psi \in L^1(\nu_\star)} G(\varphi_{k+1}, \psi)$ alternatingly. From a geometric perspective, alternating maximization corresponds to alternating projections (detailed in Appendix C.4)

$$\varphi_{k+1} = \arg\max_{\varphi \in L^1(\mu_\star)} G(\varphi, \psi_k) \implies \text{the first marginal of } \pi(\varphi_{k+1}, \psi_k) \text{ is } \mu_\star, \quad (12a)$$

$$\psi_{k+1} = \arg\max_{\psi \in L^1(\nu_\star)} G(\varphi_{k+1}, \psi) \implies \text{the second marginal of } \pi(\varphi_{k+1}, \psi_{k+1}) \text{ is } \nu_\star. \quad (12b)$$

The marginal properties of the coupling implies the Schrödinger equation (Nutz & Wiesel, 2022)

$$\varphi_\star(\mathbf{x}) = -\log \int_\Omega e^{\psi_\star(\mathbf{y}) - c_\varepsilon(\mathbf{x}, \mathbf{y})} \nu_\star(\mathrm{d}\mathbf{y}), \quad \psi_\star(\mathbf{y}) = -\log \int_\Omega e^{\varphi_\star(\mathbf{x}) - c_\varepsilon(\mathbf{x}, \mathbf{y})} \mu_\star(\mathrm{d}\mathbf{x}).$$

Since the Schrödinger potential functions $(\psi_\star, \varphi_\star)$ are not known *a priori*, the dual formulation of the static IPF algorithm was proposed to solve the alternating projections as follows:

**Static Dual IPF** : $\psi_k(\mathbf{y}) = -\log \int_\Omega e^{\varphi_k(\mathbf{x}) - c_\varepsilon(\mathbf{x}, \mathbf{y})} \mu_\star(\mathrm{d}\mathbf{x}), \quad \varphi_{k+1}(\mathbf{x}) = -\log \int_\Omega e^{\psi_k(\mathbf{y}) - c_\varepsilon(\mathbf{x}, \mathbf{y})} \nu_\star(\mathrm{d}\mathbf{y}).$

$$(13)$$

The equivalence between the primal IPF and dual IPF is further illustrated in Appendix C.5.

However, given a limited computational budget, projecting to the ideal measure $\mu_\star$ (or $\nu_\star$) in Eq.(12) at each iteration may not be practical. Instead, some close approximation $\mu_{\star, k+1}$ (or $\nu_{\star, k}$) is used at iteration $2k + 1$ (or $2k$) via Gaussian processes (Vargas et al., 2021) or neural networks (De Bortoli et al., 2021; Chen et al., 2022b). Therefore, one may resort to an approximate marginal that still achieves reasonable accuracy:

$$\mu_{2k+1} = \mu_{\star, k+1} \approx \mu_\star, \quad \nu_{2k} = \nu_{\star, k} \approx \nu_\star. \quad (14)$$

We refer to the IPF algorithm with approximate marginals as approximate IPF (aIPF) and present the static dual formulation of aIPF in Algorithm 2. The difference between IPF and aIPF is detailed in Figure 3. The structure representation (10) can be naturally extended based on approximate marginals and is also studied by Deligiannidis et al. (2021)

$$\mathrm{d}\pi_{2k} = e^{\varphi_k \oplus \psi_k - c_\varepsilon} \mathrm{d}(\mu_{\star, k} \otimes \nu_{\star, k}),$$
$$\mathrm{d}\pi_{2k-1} = e^{\varphi_k \oplus \psi_{k-1} - c_\varepsilon} \mathrm{d}(\mu_{\star, k} \otimes \nu_{\star, k-1}), \quad (15)$$

where $\pi_k$ is the approximate coupling at iteration $k$. By the structural properties in Eq.(10), the representation also applies to the dynamic settings, which involves the computation of the static IPF, followed by its integration with a diffusion bridge (Eckstein & Nutz, 2022).

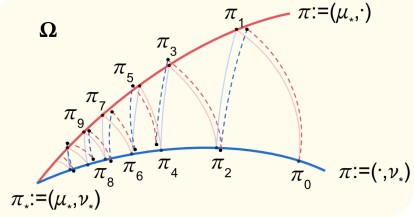

Figure 3: IPF v.s. aIPF. The approximate (or exact) projections are highlighted through the dotted (or solid) lines.

---

**Algorithm 2** One iteration of aIPF (static). The static coupling $\pi_k$ can be recovered by the structural representation in (15); the dynamic coupling $\mathbb{P}_k = \iint_{\Omega^2} \mathbb{P}_k^{\mathbf{x}_0, \mathbf{x}_T}(\cdot) \pi_k(\mathbf{x}_0, \mathbf{x}_T)$ can be solved by further learning a diffusion bridge $\mathbb{P}_k^{\mathbf{x}_0, \mathbf{x}_T}$.

---

$$\psi_k(\mathbf{y}) = -\log \int_\Omega e^{\varphi_k(\mathbf{x}) - c_\varepsilon(\mathbf{x}, \mathbf{y})} \mu_{\star,k}(\mathrm{d}\mathbf{x}), \quad \varphi_{k+1}(\mathbf{x}) = -\log \int_\Omega e^{\psi_k(\mathbf{y}) - c_\varepsilon(\mathbf{x}, \mathbf{y})} \nu_{\star,k}(\mathrm{d}\mathbf{y}). \quad (16)$$

---

### 4.3 CONVERGENCE OF COUPLINGS WITH BOUNDED DOMAIN

Despite the rich literature on the analysis of SBP within bounded domains (Chen et al., 2016), most of them are not applicable to practical scenarios where exact marginals are not available. To fill this gap, we extend the linear convergence analysis with perturbed marginals. The key to our proof is the strong convexity of the dual (11). To quantify the convergence, similar to De Bortoli (2022), we introduce an additional assumption to control the perturbation of the marginals in the sense that:

**Assumption A4** (Marginal perturbation). $U_k = \nabla \log \frac{\mathrm{d}\mu_{\star,k}}{\mathrm{d}\mathbf{x}}$ and $V_k = \nabla \log \frac{\mathrm{d}\nu_{\star,k}}{\mathrm{d}\mathbf{x}}$ are the approximate energy functions at the $k$-th iteration and are $\epsilon$-close to energy functions $U_\star$ and $V_\star$

$$\left\| U_k(\mathbf{x}) - U_\star(\mathbf{x}) \right\|_2 \le \epsilon(1 + \|\mathbf{x}\|_2), \ \ \left\| V_k(\mathbf{x}) - V_\star(\mathbf{x}) \right\|_2 \le \epsilon(1 + \|\mathbf{x}\|_2), \quad \forall \mathbf{x} \in \Omega.$$

Note that the Lipschitz cost function on $\Omega^2$ is also a standard assumption (Deligiannidis et al., 2021). It is not required here by Assumption A1, which leads to a smooth transition kernel and cost function.

Recall the connections between dynamic primal IPF and static dual IPF, we know $\epsilon$ mainly depends on the score-function $(\overrightarrow{\mathbf{z}}_t^\theta, \overleftarrow{\mathbf{z}}_t^\omega)$ estimations (Song et al., 2021a) and numerical discretizations. More concrete connections between them will be left as future work. In addition, the errors in the two marginals don't have to be the same, and we use a unified $\epsilon$ mainly for analytical convenience.

Moreover, we use the same domain $\Omega$ for both marginals to be consistent with our algorithm in Section 3. The proof can be easily extended to different domains X and Y for $\mu_\star$ and $\nu_\star$.

**Approximately linear convergence and proof sketches**  We first follow Carlier (2022); Nutz (2022); Marino & Gerolin (2020) to build a *centered* aIPF algorithm in Algorithm 3 with scaled potential functions $\bar{\varphi}_k$ and $\bar{\psi}_k$ such that $\mu_\star(\bar{\varphi}_k) = 0$. Since the summations of the potentials $\varphi_\star$ and $\psi_\star$ are unique by (10), the *centering* operation doesn't change the dual objective but ensures that the aIPF iterates are uniformly bounded in Lemma 4 with the help of the decomposition

$$\|\bar{\varphi} \oplus \bar{\psi}\|_{L^2(\mu_\star \otimes \nu_\star)}^2 = \|\bar{\varphi}\|_{L^2(\mu_\star)}^2 + \|\bar{\psi}\|_{L^2(\nu_\star)}^2 \quad \text{if} \ \ \mu_\star(\bar{\varphi}) = 0.$$

How to ensure centering with perturbed marginals in Algorithm 3 is crucial and one major novelty in our proof. We next exploit the *strong convexity* of the exponential function $e^x$ associated with the concave dual. We obtain an auxiliary result regarding the convergence of the dual and the potentials.

**Lemma 2** (Convergence of the Dual and Potentials). *Let* $(\bar{\varphi}_k, \bar{\psi}_k)_{k \ge 0}$ *be the iterates of a variant of Algorithm 2. Given assumptions A1-A4 with small enough marginal perturbations $\epsilon$, we have*

$$G(\bar{\varphi}_\star, \bar{\psi}_\star) - G(\bar{\varphi}_k, \bar{\psi}_k) \lesssim (1 - e^{-24\|c_\varepsilon\|_\infty})^k + e^{24\|c_\varepsilon\|_\infty} \epsilon,$$

$$\|\bar{\varphi}_\star - \bar{\varphi}_k\|_{L^2(\mu_\star)} + \|\bar{\psi}_\star - \bar{\psi}_k\|_{L^2(\nu_\star)} \lesssim e^{3\|c_\varepsilon\|_\infty}(1 - e^{-24\|c_\varepsilon\|_\infty})^{k/2} + e^{15\|c_\varepsilon\|_\infty} \epsilon^{1/2}.$$

Since the centering operation doesn't change the structure property (10), we are able to analyze the convergence of the static couplings. Motivated by Theorem 3 of Deligiannidis et al. (2021), we exploit the structural property (10) to estimate the $\mathbf{W}_1$ distance based on its dual formulation.

**Theorem 2** (Convergence of Static Couplings). *Given assumptions A1-A4 with small marginal perturbations $\epsilon$, the iterates of the couplings $(\pi_k)_{k \ge 0}$ in Algorithm 2 satisfy the following result*

$$\mathbf{W}_1(\pi_k, \pi_\star) \le O(e^{9\|c_\varepsilon\|_\infty}(1 - e^{-24\|c_\varepsilon\|_\infty})^{k/2} + e^{21\|c_\varepsilon\|_\infty} \epsilon^{1/2}).$$

Such a result provides the worst-case guarantee on the convergence of the static couplings $\pi_k$. For example, to obtain a $\epsilon_\star$-$\mathbf{W}_1$ distance, we can run $\Omega(e^{24\|c_\varepsilon\|_\infty}(\|c_\varepsilon\|_\infty - \log(\epsilon_\star \wedge 1)))$ iterations to

achieve the goal. Recall that $c_\varepsilon = c/\varepsilon$ (Chen et al., 2023), a large entropic-regularizer $\varepsilon$ may be needed in practice to yield reasonable performance, which also leads to specific tuning guidance on $\varepsilon$.

Our proof employs a non-geometric method to show the uniform in time stability, w.r.t. the marginals. Unlike the elegant approach (Deligiannidis et al., 2021) based on the Hilbert-Birkhoff projective metric (Chen et al., 2016), ours does not require advanced tools and may be more friendly to readers.

Recall the bridge representation in Eq.(8), we have $\mathbf{W}_1(\pi_k \otimes \mathbb{P}_k^{\mu_\star,\nu_\star}, \pi_\star \otimes \mathbb{P}_\star^{\mu_\star,\nu_\star}) \le \mathbf{W}_1(\pi_k, \pi_\star) + \mathbf{W}_1(\mathbb{P}_k^{\mu_\star,\nu_\star}, \mathbb{P}_\star^{\mu_\star,\nu_\star})$. Assume the same assumptions as in Theorem 2, we arrive at the final result:

**Proposition 2** (Convergence of Dynamic Couplings). *The iterates of the dynamic couplings* $(\mathbb{P}_k)_{k \ge 0}$ *in Algorithm 1 satisfy the following result*

$$\mathbf{W}_1(\mathbb{P}_k, \mathbb{P}_\star) \le O(e^{9\|c_\varepsilon\|_\infty}(1 - e^{-24\|c_\varepsilon\|_\infty})^{k/2} + e^{21\|c_\varepsilon\|_\infty}\epsilon^{1/2}) + \mathbf{W}_1(\mathbb{P}_k^{\mu_\star,\nu_\star}, \mathbb{P}_\star^{\mu_\star,\nu_\star}).$$

The result paves the way for understanding the general convergence of the dynamic IPF algorithm by incorporating a proper approximation of the diffusion bridge (Heng et al., 2022).

## 5 EMPIRICAL SIMULATIONS

### 5.1 GENERATION OF 2D SYNTHETIC DATA

We first employ the reflected SB algorithm to generate three synthetic examples: checkerboard and Gaussian mixtures from a Gaussian prior and spiral from a moon prior. The domains are defined to be flower, octagon, and heart, where all boundary points are defined to have proper unit-vectors. We follow Chen et al. (2022b) and adopt a U-net to model $(\overrightarrow{\mathbf{z}}_t^\theta, \overleftarrow{\mathbf{z}}_t^\omega)$. We chose RVP-SDE as the base simulator from time 0 to $T = 1$, where the dynamics are discretized into 100 steps.

Our generated examples are presented in Figure 1 and 4. We see that all the data are generated smoothly from the prior and the forward and backward process matches with each other elegantly. To the best of our knowledge, this is the first algorithm (with OT guarantees) that works on custom domains. Other related work, such as Lou & Ermon (2023), mainly focuses on hypercubes in computer vision. We also visualize the forward-backward policies $\overleftarrow{\mathbf{z}}_t^\omega$ and $\overrightarrow{\mathbf{z}}_t^\theta$ in Figure 4. Our observations reveal that the forward vector fields $\overrightarrow{\mathbf{z}}_t^\theta$ demonstrate substantial nonlinearity when compared to the linear forward policy in SGMs, and furthermore, the forward vector fields exhibit pronounced dissimilarity when compared to the backward vector fields $\overleftarrow{\mathbf{z}}_t^\omega$.

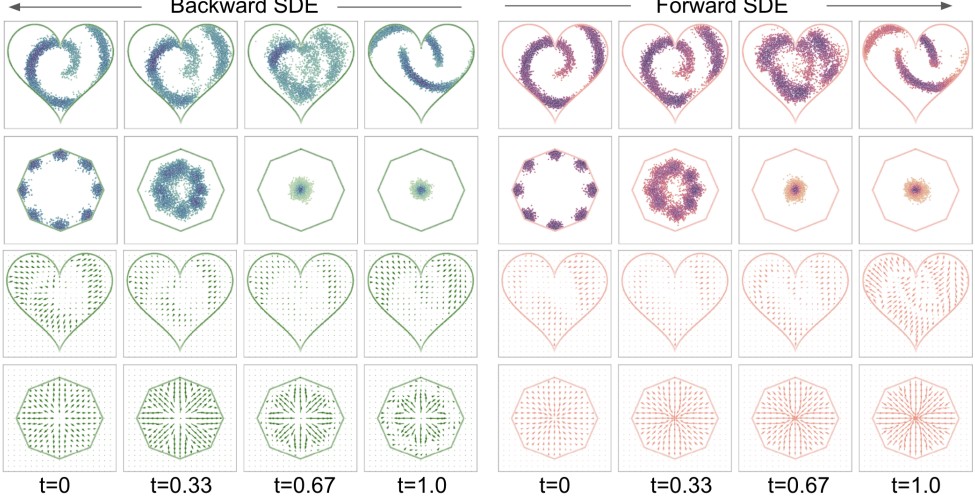

Figure 4: Demo of generative samples (top) and vector fields (bottom) based on Reflected SB.

### 5.2 GENERATION OF IMAGE DATA

We test our method on large-scale image datasets using CIFAR-10 and ImageNet 64×64. As the RGB value is between $[0, 1]$, we naturally select the domain as $\Omega = [0, 1]^d$, where $d = 3 \times 32 \times 32$

for the CIFAR-10 task and $d = 3 \times 64 \times 64$ for the ImageNet task. It is known that the SB system can be initialized with score-based generative models (Chen et al., 2022b) and the warm-up study for reflected SB is presented in Appendix D.2. We choose RVE-SDE as the prior path measure. The prior distribution of $\nu_\star$ is the uniform distribution on $\Omega$. The SDE is discretized into 1000 steps. In both scenarios, images are generated unconditionally, and the quality of the samples is evaluated using Frechet Inception Distance (FID) over 50,000 samples. The forward score function is modeled using U-net structure; the backward score function uses NCSN++ (Song et al., 2021b) for the CIFAR-10 task and ADM (Dhariwal & Nichol, 2022) for the ImageNet task. Details of the experiments are shown in Appendix D.

We have included baselines for both constrained and unconstrained generative models and summarized the experimental results in Table 1. While our model may not surpass the state-of-the-art models, the minor improvement over the unconstrained SB-FBSDE (Chen et al., 2022b) underscores the effectiveness of the reflection operation. Moreover, the experiments verify the scalability of the reflected model and the training process is consistent with the findings in Lou & Ermon (2023), where the reflection in cube domains is easy to implement and the generation becomes more stable. Sample outputs

| CIFAR-10 | Constrained | OT | NLL | FID |
|---|---|---|---|---|
| MCSN++ (Song et al., 2021b) | No | No | 2.99 | 2.20 |
| DDPM (Ho et al., 2020) | No | No | 3.75 | 3.17 |
| SB-FBSDE (Chen et al., 2022b) | No | Yes | - | 3.01 |
| Reflected SGM (Lou & Ermon, 2023) | Yes | No | 2.68 | 2.72 |
| **Ours** | Yes | Yes | 3.08 | 2.98 |
| | | | | |
| ImageNet 64×64 | | | | |
| PGMGAN (Armandpour et al., 2021) | No | No | – | 21.73 |
| GLIDE (Li et al., 2023) | No | No | – | 29.18 |
| GRB (Park & Shin, 2022) | No | No | – | 26.57 |
| **Ours** | Yes | Yes | 3.20 | 23.95 |

Table 1: Evaluation of generative models on image data.

are showcased in Figure 5 (including MNIST), with additional figures available in Appendix D. Notably, our generated samples exhibit diversity and are visually indistinguishable from real data.

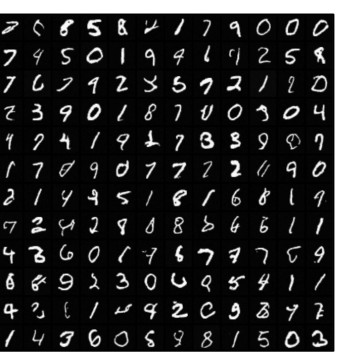 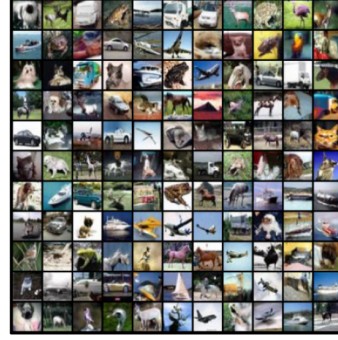 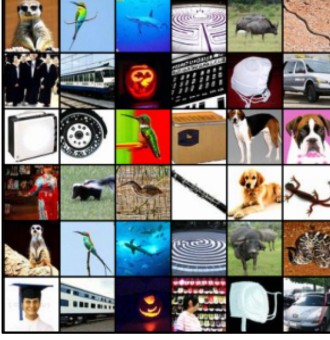

Figure 5: Samples via reflected SB on MNIST (left), CIFAR10 (middle), and ImageNet 64 (right).

## 6  CONCLUSION

Reflected diffusion models, which are motivated by thresholding techniques, introduce explicit score-matching loss through reflected Brownian motion. Traditionally, these models are applied to hypercube-related domains and necessitate specific diffeomorphic mappings for extension to other domains. To enhance generality with optimal transport guarantees, we introduce the Reflected Schrödinger Bridge, which employs reflected forward-backward stochastic differential equations with Neumann and Robin boundary conditions. We establish connections between dynamic and static IPF algorithms in both primal and dual formulations. Additionally, we provide an approximate linear convergence analysis of the dual, potential, and couplings to deepen our understanding of the dynamic IPF algorithm. Empirically, our algorithm can be applied to any smooth domains using RVE-SDE and RVP-SDE. We evaluate its performance on 2D synthetic examples and standard image benchmarks, underscoring its competitiveness in constrained generative modeling.

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

## A    RELATED WORKS

**Constrained Sampling**    Bubeck et al. (2018) studied the convergence of Langevin Monte Carlo within bounded domains. His work revealed a polynomial sample time for log-concave distributions, which is later extended to non-convex settings by Lamperski (2021). Furthermore, the exploration of constrained sampling in challenging scenarios with ill-conditioned and non-smooth distributions was explored by Kook et al. (2022), who leveraged Hamiltonian Monte Carlo techniques. Other constrained sampling works include proximal Langevin dynamics (Brosse et al., 2017) and mirrored Langevin dynamics (Hsieh et al., 2018).

**Constrained Generation**    De Bortoli et al. (2022); Huang et al. (2022) studied the extension of diffusion models on Riemannian manifolds, and the convergence is further analyzed by De Bortoli (2022). This groundwork subsequently motivated follow-up research, including implicit score-matching loss via log-barrier methods and reflected Brownian motion (Fishman et al., 2023) and Schrödinger bridge (Thornton et al., 2022) on the Riemannian manifold. Alternatively, drawing inspiration from the popular thresholding technique in real-world diffusion applications, Lou & Ermon (2023) proposed to train explicit score-matching loss based on reflected Brownian motion, which demonstrated compelling empirical performance. Liu et al. (2023) employed Doob's h-transform to learn diffusion bridges on various constrained domains. The study of reflected Schrödinger bridge was initiated by Caluya & Halder (2021) in the control community and has shown remarkable performance in low-dimensional problems.

**Notations**    $\Omega$ is the bounded domain of interest, $\partial\Omega$ denotes the boundary, and $D$ is the radius of a ball centered at the origin that covers $\Omega$. $\nabla\cdot$ and $\nabla$ denote the divergence and gradient operator (with respect to $\mathbf{x}$). $\varepsilon$ is the entropic regularizer; $\epsilon$ controls the perturbation of the marginals. $c_\varepsilon$ and $c$ are cost functions in EOT, and $c_\varepsilon = c/\varepsilon$. $\psi_\star$ and $\varphi_\star$ are the Schrödinger potentials; $\nabla\log\overrightarrow{\psi}(\mathbf{x}_t, t)$ and $\nabla\log\overleftarrow{\varphi}(\mathbf{x}_t, t)$ are the forward-backward score functions in reflected FB-SDE. $\mathcal{D}(\mu_\star, \nu_\star) \subset C(\Omega, [0, T])$ is the path space with marginals $\mu_\star$ (data distribution) at time $t = 0$ and $\nu_\star$ (prior distribution) at $t = T$, $\Pi(\mu_\star, \nu_\star) \subset \Omega^2$ is the product space containing couplings with the first marginal $\mu_\star$ and second marginal $\nu_\star$.

## B    REFLECTED FORWARD-BACKWARD SDE

### B.1    FROM REFLECTED SCHRÖDINGER BRIDGE TO REFLECTED FB-SDE

We consider the stochastic control of the reflected SBP

$$
\begin{aligned}
&\inf_{\boldsymbol{u}\in\mathcal{U}} \mathbb{E}\left\{ \int_0^T \frac{1}{2}\|\boldsymbol{u}(\mathbf{x}, t)\|_2^2 \mathrm{d}t \right\} \\
&\text{s.t. } \mathrm{d}\mathbf{x}_t = [\boldsymbol{f}(\mathbf{x}, t) + g(t)\boldsymbol{u}(\mathbf{x}, t)]\,\mathrm{d}t + \sqrt{2\varepsilon}g(t)\mathrm{d}\mathbf{w}_t + \mathbf{n}(\mathbf{x})\mathrm{d}\mathbf{L}_t \\
&\qquad \mathbf{x}_0 \sim \mu_\star, \ \ \mathbf{x}_T \sim \nu_\star, \ \ \mathbf{x}_t \in \Omega,
\end{aligned}
\tag{17}
$$

where $\Omega$ is the state-space of $\mathbf{x}$ and $\boldsymbol{u} : \Omega \times [0, T] \to \mathbb{R}^d$ is the control variable in the space of $\mathcal{U}$; $\boldsymbol{f} : \Omega \times [0, T] \to \mathbb{R}^d$ is the vector field; $\mathbf{w}_t$ denotes the Brownian motion; The expectation is evaluated w.r.t the PDF $\rho(\mathbf{x}, t)$ of (17); $\varepsilon$ is the diffusion term and also the entropic regularizer; $\mathbf{L}$ is the local time supported on $\{t \in [0, T] | \mathbf{x}_t \in \partial\Omega\}$ and forces the particle to go back to $\Omega$. More precisely, $\mathbf{L}$ is a continuous non-decreasing process with $\mathbf{L}_0 = 0$ and it increases only when $\mathbf{x}_t$ hits the boundary $\partial\Omega$, that is,

$$
\mathbf{L}_t = \int_0^t \mathbf{1}_{\{\mathbf{x}_s \in \partial\Omega\}} d\mathbf{L}_s.
\tag{18}
$$

The existence and uniqueness of SDE (17) can be addressed through the so-called *Skorokhod problem* (Skorokhod, 1961) which amounts to finding the decomposition for any given continuous path $\mathbf{w}_t \in C(\mathbb{R}^d, [0, T])$, there exists a pair $(\mathbf{y}_t, \mathbf{L}_t)$ such that

$$
\mathbf{w}_t = \mathbf{y}_t + \mathbf{L}_t,
$$

where $\mathbf{y}_t \in C(\Omega, [0, T])$ and $\mathbf{L}_t$ satisfies (18) (Lions & Sznitman, 1984).

Rewrite reflected SBP into a variational form (Chen et al., 2021)

$$\inf_{\boldsymbol{u} \in \mathcal{U}, \rho} \int_0^T \int_\Omega \frac{1}{2} \|\boldsymbol{u}(\mathbf{x}, t)\|_2^2 \rho(\mathbf{x}, t) \mathrm{d}\mathbf{x} \mathrm{d}t \tag{19}$$

$$\text{s.t.} \ \frac{\partial \rho}{\partial t} + \nabla \cdot \mathbf{J}|_{\mathbf{x} \in \Omega} = 0, \ \langle \mathbf{J}, \mathbf{n} \rangle|_{\mathbf{x} \in \partial \Omega} = 0, \tag{20}$$

where $\mathbf{J}$ is the probability flux of continuity equation (Pavliotis, 2014) given by

$$\mathbf{J} \equiv \rho(\boldsymbol{f} + g\boldsymbol{u}) - \varepsilon g^2 \nabla \rho. \tag{21}$$

Explore the Lagrangian of (19) and incorporate a multiplier: $\phi(\mathbf{x}, t) : \Omega \times [0, T] \to \mathbb{R}$

$$\mathcal{L}(\rho, \boldsymbol{u}, \phi) = \int_0^T \int_\Omega \frac{1}{2} \|\boldsymbol{u}\|_2^2 \rho + \phi\left(\frac{\partial \rho}{\partial t} + \nabla \cdot \mathbf{J}\right) \mathrm{d}\mathbf{x} \mathrm{d}t \tag{22}$$

$$= \int_0^T \int_\Omega \left(\frac{1}{2} \rho \|\boldsymbol{u}\|_2^2 - \rho \frac{\partial \phi}{\partial t} - \langle \nabla \phi, \mathbf{J} \rangle\right) \mathrm{d}\mathbf{x} \mathrm{d}t + \underbrace{\int_\Omega \phi \rho|_{t=0}^T \mathrm{d}\mathbf{x}}_{\text{constant term}} + \underbrace{\int_0^T \int_{\partial \Omega} \langle \mathbf{J}, \mathbf{n} \rangle \mathrm{d}\sigma(\mathbf{x}) \mathrm{d}t}_{:= 0 \text{ by Eq.(20)}},$$

where the second equation follows by Stokes' theorem.

Plugging (21) into (22) and ignoring constant terms, we have

$$\overline{\mathcal{L}}(\rho, \boldsymbol{u}, \phi) = \int_0^T \int_\Omega \left(\frac{1}{2} \rho \|\boldsymbol{u}\|_2^2 - \rho \frac{\partial \phi}{\partial t} - \langle \nabla \phi, \rho(\boldsymbol{f} + g\boldsymbol{u}) - \varepsilon g^2 \nabla \rho \rangle\right) \mathrm{d}\mathbf{x} \mathrm{d}t. \tag{23}$$

The optimal control $\boldsymbol{u}^\star$ follows by taking gradient with respect to $\boldsymbol{u}$

$$\boldsymbol{u}^\star(\mathbf{x}, t) = g(t) \nabla \phi(\mathbf{x}, t). \tag{24}$$

Plugging $\boldsymbol{u}^\star(\mathbf{x}, t)$ into (23) and setting $\overline{\mathcal{L}}(\rho, \boldsymbol{u}^\star, \phi) \equiv 0$, we apply integration by parts and derive

$$0 = -\int_0^T \int_\Omega \left(\frac{1}{2} \rho g^2 \|\nabla \phi\|_2^2 + \rho \frac{\partial \phi}{\partial t} + \rho \langle \nabla \phi, \boldsymbol{f} \rangle - \varepsilon g^2 \langle \nabla \phi, \nabla \rho \rangle\right) \mathrm{d}\mathbf{x} \mathrm{d}t$$

$$= -\int_0^T \int_\Omega \rho\left(\frac{1}{2} g^2 \|\nabla \phi\|_2^2 + \frac{\partial \phi}{\partial t} + \langle \nabla \phi, \boldsymbol{f} \rangle + \varepsilon g^2 \Delta \phi\right) \mathrm{d}\mathbf{x} \mathrm{d}t + \int_0^T \int_{\partial \Omega} \varepsilon g^2 \rho \langle \nabla \phi, \mathbf{n} \rangle \mathrm{d}\sigma(\mathbf{x}) \mathrm{d}t.$$

This yields the following constrained *Hamilton–Jacobi–Bellman* (HJB) PDE:

$$\begin{cases} \frac{\partial \phi}{\partial t} + \varepsilon g^2 \Delta \phi + \langle \nabla \phi, \boldsymbol{f} \rangle = -\frac{1}{2} \|g(t) \nabla \phi(\mathbf{x}, t)\|_2^2 & \text{in } \Omega \\ \langle \nabla \phi, \mathbf{n} \rangle = 0 & \text{on } \partial \Omega. \end{cases}$$

Applying the Cole-Hopf transformation:

$$\overrightarrow{\psi}(\mathbf{x}, t) = \exp\left(\frac{\phi(\mathbf{x}, t)}{2\varepsilon}\right), \ \phi(\mathbf{x}, t) = 2\varepsilon \log \overrightarrow{\psi}(\mathbf{x}, t), \tag{25}$$

then we see that $\overrightarrow{\psi}$ satisfies a backward Kolmogorov equation with Neumann boundary condition

$$\begin{cases} \frac{\partial \overrightarrow{\psi}}{\partial t} + \varepsilon g^2 \Delta \overrightarrow{\psi} + \langle \nabla \overrightarrow{\psi}, \boldsymbol{f} \rangle = 0 & \text{in } \Omega \\ \langle \nabla \overrightarrow{\psi}, \mathbf{n} \rangle = 0 & \text{on } \partial \Omega. \end{cases}$$

On the other hand, we set

$$\overleftarrow{\varphi}(\mathbf{x}, t) = \rho^\star(\mathbf{x}, t) / \overrightarrow{\psi}(\mathbf{x}, t), \tag{26}$$

where $\rho^\star(\mathbf{x}, t)$ is the probability density of Eq.(19) given the optimal control variable $\boldsymbol{u}^\star$. Then from $\rho^\star = \overleftarrow{\varphi}\,\overrightarrow{\psi}$, Eq.(20) can be further simplified to

$$
\begin{aligned}
0 &= \partial_t \rho^\star + \nabla \cdot \left[ \rho^\star \big(\boldsymbol{f} + g\boldsymbol{u}^\star\big) - \varepsilon g^2 \nabla \rho^\star \right] \\
&= \partial_t (\overleftarrow{\varphi}\,\overrightarrow{\psi}) + \nabla \cdot \left[ \overleftarrow{\varphi}\,\overrightarrow{\psi}\,\big(\boldsymbol{f} + g^2 \nabla \phi\big) - \varepsilon g^2 \nabla(\overleftarrow{\varphi}\,\overrightarrow{\psi}) \right] \\
&= (\partial_t \overleftarrow{\varphi})\overrightarrow{\psi} + \overleftarrow{\varphi}(\partial_t \overrightarrow{\psi}) + \nabla \cdot \left[ \overleftarrow{\varphi}\,\overrightarrow{\psi}\,\big(\boldsymbol{f} + 2\varepsilon g^2 \nabla \log \overrightarrow{\psi}\big) \right] - \varepsilon g^2 \Delta(\overleftarrow{\varphi}\,\overrightarrow{\psi}) \\
&= \cdots = \overrightarrow{\psi}\left( \partial_t \overleftarrow{\varphi} + \nabla \cdot (\overleftarrow{\varphi}\,\boldsymbol{f} - \varepsilon g^2 \nabla \overleftarrow{\varphi}) \right),
\end{aligned}
$$

where we use the identify $\Delta(\overrightarrow{\psi}\,\overleftarrow{\varphi}) = \overleftarrow{\varphi}\Delta\overrightarrow{\psi} + \Delta\overleftarrow{\varphi}\,\overrightarrow{\psi} + 2\langle \nabla\overrightarrow{\psi}, \nabla\overleftarrow{\varphi} \rangle$. Then we arrive at the forward Kolmogorov equation with the Robin boundary condition

$$
\begin{cases}
\partial_t \overleftarrow{\varphi} + \nabla \cdot \big(\overleftarrow{\varphi}\,\boldsymbol{f} - \varepsilon g^2 \nabla \overleftarrow{\varphi}\big) = 0 & \text{in } \Omega \\
\langle \overleftarrow{\varphi}\,\boldsymbol{f} - \varepsilon g^2 \nabla \overleftarrow{\varphi}, \mathbf{n} \rangle = 0 & \text{on } \partial\Omega,
\end{cases}
$$

where the second boundary condition follows by invoking the Stokes' theorem for the first equation.

Plugging Eq.(24) and Eq.(25) into Eq.(17), the backward PDE corresponds to the forward SDE

$$
\mathrm{d}\mathbf{x}_t = \left[ \boldsymbol{f}(\mathbf{x}_t, t) + 2\varepsilon g(t)^2 \nabla \log \overrightarrow{\psi}(\mathbf{x}_t, t) \right] \mathrm{d}t + \sqrt{2\varepsilon} g(t) \mathrm{d}\mathbf{w}_t + \mathbf{n}(\mathbf{x}) \mathrm{d}\mathbf{L}_t, \quad \mathbf{x}_0 \sim \mu_\star.
$$

Reversing the forward SDE (Williams, 1987; Cattiaux, 1988) with $\log \overrightarrow{\psi}(\cdot, t) + \log \overleftarrow{\varphi}(\cdot, t) = \log \rho^\star(\cdot, t)$ based on Eq.(26), we arrive at the backward SDE

$$
\mathrm{d}\mathbf{x}_t = \left[ \boldsymbol{f}(\mathbf{x}_t, t) - 2\varepsilon g(t)^2 \nabla \log \overleftarrow{\varphi}(\mathbf{x}_t, t) \right] \mathrm{d}t + \sqrt{2\varepsilon} g(t) \mathrm{d}\overline{\mathbf{w}}_t + \mathbf{n}(\mathbf{x}) \mathrm{d}\overline{\mathbf{L}}_t, \quad \mathbf{x}_T \sim \nu_\star.
$$

Our derivation is in a spirit similar to Caluya & Halder (2021). The difference is that the proof is derived from the perspective of probability flux and enables us to derive the Neunman and Robin boundaries more explicitly.

**Remark 1.** *Regarding the scores $\nabla \log \overrightarrow{\psi}$ and $\nabla \log \overleftarrow{\varphi}$ at $t = 0$ and $T$, we follow the standard truncation techniques (Ho et al., 2020; Fishman et al., 2023) and fix them to $\mathbf{0}$. We refer readers to Appendix C of Song et al. (2021b) and Appendix B of Song et al. (2021a) for more discussions.*

### B.2 CONNECTIONS BETWEEN REFLECTED FB-SDEs AND FLOW-BASED MODELS

Similar to Fishman et al. (2023); Lou & Ermon (2023), our flow representation in Eq.(20) together with (26) naturally yields

**Proposition 3** (Probability Flow ODE). *Consider the reflected FB-SDEs (6) with Neumann and Robin boundary conditions. The corresponding probability flow ODE is given by*

$$
\mathrm{d}\mathbf{x}_t = \left[ \boldsymbol{f}(\mathbf{x}_t, t) + \varepsilon g(t)^2 \big(\nabla \log \overrightarrow{\psi}(\mathbf{x}_t, t) - \nabla \log \overleftarrow{\varphi}(\mathbf{x}_t, t)\big) \right] \mathrm{d}t.
$$

The result is the same as in Chen et al. (2022b) and provides a stable alternative to compute the log-likelihood of the data.

## C CONVERGENCE OF DUAL, POTENTIALS, AND COUPLINGS

Next, we modify Algorithm 2 following the centering method developed in (Carlier, 2022).

The algorithm differs from Algorithm 2 in that an additional centering operation is included in the updates of $\bar{\varphi}_{k+1}$ to ensure $\mu_\star(\bar{\varphi}_{k+1}) = 0$. Notably, $\mu_\star$ is required for the centering operation to upper bound the divergence, although it is not directly accessible and no implementation is needed. The main contribution of the centering operation is that the two coordinates $(\bar{\varphi}, \bar{\psi})$ become separable

$$
\|\bar{\varphi} \oplus \bar{\psi}\|_{L^2(\mu_\star \otimes \nu_\star)}^2 = \|\bar{\varphi}\|_{L^2(\mu_\star)}^2 + \|\bar{\psi}\|_{L^2(\nu_\star)}^2 \quad \text{if} \quad \mu_\star(\bar{\varphi}) = 0. \tag{29}
$$

---

**Algorithm 3** Centered Sinkhorn. Set $\bar{\varphi}_0 := 0$. For $k \geq 0$, the iterate follows

$$\bar{\psi}_k(\mathbf{y}) := -\log \int_\Omega e^{\bar{\varphi}_k(\mathbf{x}) - c_\varepsilon(\mathbf{x}, \mathbf{y})} \mu_{\star,k}(\mathrm{d}\mathbf{x}) \tag{27}$$

$$\bar{\varphi}_{k+1}(\mathbf{x}) := -\log \int_\Omega e^{\bar{\psi}_k(\mathbf{y}) - c_\varepsilon(\mathbf{x}, \mathbf{y})} \nu_{\star,k}(\mathrm{d}\mathbf{y}) + \lambda_k, \quad \text{where} \tag{28}$$

$$\lambda_k := \int_\Omega \log\left(\int_\Omega e^{\bar{\psi}_k(\mathbf{y}) - c_\varepsilon(\mathbf{x}, \mathbf{y})} \nu_{\star,k}(\mathrm{d}\mathbf{y})\right) \mu_\star(\mathrm{d}\mathbf{x}).$$

---

The coordinate ascent is equivalent to the following updates

$$\bar{\psi}_k(y) = \underset{\bar{\psi} \in L^1(\nu_\star)}{\arg\max}\, G(\bar{\varphi}_k, \bar{\psi}), \quad \bar{\varphi}_k(y) = \underset{\bar{\varphi} \in L^1(\mu_\star): \mu_\star(\bar{\varphi})=0}{\arg\max}\, G(\bar{\varphi}, \bar{\psi}_k).$$

The relation between the Schrödinger potentials $(\varphi_k, \psi_k)$ and centered Schrödinger potentials $(\bar{\varphi}_k, \bar{\psi}_k)$ is characterized as follows

**Lemma 3.** *Denote by $(\varphi_k, \psi_k)$ the Sinkhorn iterates in Algorithm 2. For all $k \geq 0$, $\mu_\star(\varphi_k) = -(\lambda_0 + \cdots + \lambda_{k-1})$. Moreover, we have*

$$\bar{\varphi}_k = \varphi_k - \mu_\star(\varphi_k), \quad \bar{\psi}_k = \psi_k + \mu_\star(\psi_k).$$

*In particular, $\bar{\varphi}_k \oplus \bar{\psi}_k = \varphi_k \oplus \psi_k$ and $G(\bar{\varphi}_k, \bar{\psi}_k) = G(\varphi_k, \psi_k)$.*

**Proof** Applying the induction method completes the proof directly. $\qquad \square$

Recall how $\bar{\psi}_k$ is defined through the Schrödinger equation

The second marginal of $\pi_{2k}(\bar{\varphi}_k, \bar{\psi}_k) = e^{\bar{\varphi}_k \oplus \bar{\psi}_k - c_\varepsilon} \mathrm{d}(\mu_{\star,k} \otimes \nu_{\star,k})$ is $\nu_{\star,k}$,

as in Eq.(14). However, $\mathrm{d}\pi_{2k+1}(\bar{\varphi}_{k+1}, \bar{\psi}_k) = e^{\bar{\varphi}_{k+1} \oplus \bar{\psi}_k - c_\varepsilon} \mathrm{d}(\mu_{\star,k} \otimes \nu_{\star,k})$ fails to yield the first marginal $\mu_{\star,k}$ due to the centering constraint.

Next, we show the modified iterates are bounded by the cost function $c$.

**Lemma 4.** *For every $k \geq 0$, the potentials are bounded by*

$$\|\bar{\varphi}_k\|_\infty \leq 2\|c_\varepsilon\|_\infty, \quad \|\bar{\psi}_k\|_\infty \leq 3\|c_\varepsilon\|_\infty.$$

**Proof** By Assumption A1, the transition kernel $\mathcal{K}(\mathbf{x}, \mathbf{y}) = e^{-c_\varepsilon(\mathbf{x}, \mathbf{y})}$ associated with $\mathbf{x}_t = \boldsymbol{f}(\mathbf{x}_t, t)\mathrm{d}t + \sqrt{2\varepsilon}g(t)\mathrm{d}\mathbf{w}_t + \mathbf{n}(\mathbf{x})\mathrm{d}\mathbf{L}_t$ is smooth in $\Omega$, hence the cost function is Lipschitz continuous, which implies the cost function is bounded (denoted by a constant $c_\varepsilon$).

Recall the definition of $\bar{\varphi}_{k+1}$ in Algorithm 3, we have $\forall \mathbf{x}_1, \mathbf{x}_2 \in \Omega$,

$$\bar{\varphi}_{k+1}(\mathbf{x}_1) - \bar{\varphi}_{k+1}(\mathbf{x}_2)$$
$$= \log \int_\Omega e^{\bar{\psi}_k(\mathbf{y}) - c_\varepsilon(\mathbf{x}_2, \mathbf{y})} \nu_{\star,k}(\mathrm{d}\mathbf{y}) - \log \int_\Omega e^{\bar{\psi}_k(\mathbf{y}) - c_\varepsilon(\mathbf{x}_1, \mathbf{y})} \nu_{\star,k}(\mathrm{d}\mathbf{y})$$
$$\leq \log\left[e^{\sup_{\mathbf{y} \in \Omega} |c_\varepsilon(\mathbf{x}_1, \mathbf{y}) - c_\varepsilon(\mathbf{x}_2, \mathbf{y})|} \int_\Omega e^{\bar{\psi}_k(\mathbf{y}) - c_\varepsilon(\mathbf{x}_1, \mathbf{y})} \nu_{\star,k}(\mathrm{d}\mathbf{y})\right] - \log \int_\Omega e^{\bar{\psi}_k(\mathbf{y}) - c_\varepsilon(\mathbf{x}_1, \mathbf{y})} \nu_{\star,k}(\mathrm{d}\mathbf{y})$$
$$= \sup_{\mathbf{y} \in \Omega} |c_\varepsilon(\mathbf{x}_1, \mathbf{y}) - c_\varepsilon(\mathbf{x}_2, \mathbf{y})| \leq 2\|c_\varepsilon\|_\infty.$$

As $\mu_\star(\bar{\varphi}_k) = 0$, we have $\sup_x \bar{\varphi}_k(\mathbf{x}) \geq 0$ and $\inf_{\mathbf{x}} \bar{\varphi}_k(\mathbf{x}) \leq 0$, hence the above implies $\|\bar{\varphi}_k\|_\infty \leq 2\|c_\varepsilon\|_\infty$. The definition of $\bar{\psi}_k$ in Eq.(27) yields $\|\bar{\psi}_k\|_\infty \leq \|\bar{\varphi}_k\|_\infty + \|c_\varepsilon\|_\infty \leq 3\|c_\varepsilon\|_\infty$. $\qquad \square$

The key to the proof is to adopt the strong convexity of the function $e^x$ for $x \in [-\alpha, \infty)$ and some constant $\alpha \in \mathbb{R}$,

$$e^b - e^a \geq (b - a)e^a + \frac{e^{-\alpha}}{2}|b - a|^2 \quad \text{for } a, b \in [-\alpha, \infty). \tag{30}$$

We also present two supporting lemmas in order to complete the proof

**Lemma 5.** *Given $\varphi, \varphi' \in L^2(\mu_\star)$ and $\psi, \psi' \in L^2(\nu_\star)$, and define*

$$\partial_1 G(\varphi, \psi)(\mathbf{x}) = 1 - \int_\Omega e^{\varphi(\mathbf{x}) + \psi(\mathbf{y}) - c_\varepsilon(\mathbf{x}, \mathbf{y})} \nu_\star(\mathrm{d}\mathbf{y})$$

$$\partial_2 G(\varphi, \psi)(\mathbf{y}) = 1 - \int_\Omega e^{\varphi(\mathbf{x}) + \psi(\mathbf{y}) - c_\varepsilon(\mathbf{x}, \mathbf{y})} \mu_\star(\mathrm{d}\mathbf{x}). \tag{31}$$

*If both $\varphi \otimes \psi - c_\varepsilon \geq -\alpha$ and $\varphi' \oplus \psi' - c_\varepsilon \geq -\alpha$ for some $\alpha \in \mathbb{R}$, we have*

$$G(\varphi', \psi') - G(\varphi, \psi) \geq \int_\Omega \partial_1 G(\varphi', \psi')(\mathbf{x})[\varphi'(\mathbf{x}) - \varphi(\mathbf{x})]\mu_\star(\mathrm{d}\mathbf{x})$$

$$+ \int_\Omega \partial_2 G(\varphi', \psi')(\mathbf{y})[\psi'(\mathbf{y}) - \psi(\mathbf{y})]\nu_\star(\mathrm{d}\mathbf{y})$$

$$+ \frac{e^{-\alpha}}{2} \|(\varphi - \varphi') \oplus (\psi - \psi')\|_{L^2(\mu_\star \otimes \nu_\star)}.$$

**Proof** By Eq.(30), we have

$$G(\varphi', \psi') - G(\varphi, \psi)$$

$$= \mu_\star(\varphi' - \varphi) + \nu_\star(\psi' - \psi) + \iint_{\Omega^2} (e^{\varphi \oplus \psi - c_\varepsilon} - e^{\varphi' \oplus \psi' - c_\varepsilon})\mathrm{d}(\mu_\star \otimes \nu_\star)$$

$$\geq \mu_\star(\varphi' - \varphi) + \nu_\star(\psi' - \psi) + \iint_{\Omega^2} (\varphi \oplus \psi - \varphi' \oplus \psi')e^{\varphi' \oplus \psi' - c_\varepsilon}\mathrm{d}(\mu_\star \otimes \nu_\star)$$

$$+ \frac{e^{-\alpha}}{2} \iint_{\Omega^2} \|\varphi \oplus \psi - \varphi' \oplus \psi'\|_2^2 \mathrm{d}(\mu_\star \otimes \nu_\star)$$

$$= \int_\Omega \partial_1 G(\varphi', \psi')(\mathbf{x})[\varphi'(x) - \varphi(\mathbf{x})]\mu_\star(\mathrm{d}\mathbf{x}) + \int_\Omega \partial_2 G(\varphi', \psi')(\mathbf{y})[\psi'(\mathbf{y}) - \psi(\mathbf{y})]\nu_\star(\mathrm{d}\mathbf{y})$$

$$+ \frac{e^{-\alpha}}{2} \|(\varphi - \varphi') \oplus (\psi - \psi')\|_{L^2(\mu_\star \otimes \nu_\star)}.$$

$\square$

**Lemma 6.** *Given a small $\epsilon \leq \frac{1}{(D+1)^2}$, we have*

$$G(\bar{\varphi}_{k+1}, \bar{\psi}_{k+1}) - G(\bar{\varphi}_k, \bar{\psi}_k) \geq \frac{\sigma}{2} \left( \|\bar{\varphi}_{k+1} - \bar{\varphi}_k\|_{L^2(\mu_\star)}^2 + \|\bar{\psi}_{k+1} - \bar{\psi}_k\|_{L^2(\nu_\star)}^2 \right) - O(\epsilon),$$

*where $\sigma := e^{-6\|c_\varepsilon\|_\infty}$; the big-O notation mainly depends on volume of the domain $\Omega$.*

**Proof** We first decompose the LHS as follows

$$G(\bar{\varphi}_{k+1}, \bar{\psi}_{k+1}) - G(\bar{\varphi}_k, \bar{\psi}_k) = \underbrace{G(\bar{\varphi}_{k+1}, \bar{\psi}_{k+1}) - G(\bar{\varphi}_{k+1}, \bar{\psi}_k)}_{\text{I}} + \underbrace{G(\bar{\varphi}_{k+1}, \bar{\psi}_k) - G(\bar{\varphi}_k, \bar{\psi}_k)}_{\text{II}}.$$

For the estimate of I, by Lemma 5 with $\sigma = e^{-6\|c_\varepsilon\|_\infty}$, we have

$$\text{I} \geq \int_\Omega \partial_2 G(\bar{\varphi}_{k+1}, \bar{\psi}_{k+1})(\mathbf{y})[\bar{\psi}_{k+1}(\mathbf{y}) - \bar{\psi}_k(\mathbf{y})]\nu_\star(\mathrm{d}\mathbf{y}) + \frac{\sigma}{2}\|\bar{\psi}_k - \bar{\psi}_{k+1}\|_{L^2(\nu_\star)}.$$

For the integral above, by the definition of $\partial_2 G$ in Eq.(31), we have

$$\partial_2 G(\bar{\varphi}_{k+1}, \bar{\psi}_{k+1})(\mathbf{y})\nu_\star(\mathrm{d}\mathbf{y})$$

$$= \nu_\star(\mathrm{d}\mathbf{y}) - \int_\Omega e^{\bar{\varphi}_{k+1}(\mathbf{x}) + \bar{\psi}_{k+1}(\mathbf{y}) - c_\varepsilon(\mathbf{x}, \mathbf{y})}\mu_\star(\mathrm{d}\mathbf{x})\nu_\star(\mathrm{d}\mathbf{y})$$

$$= \nu_\star(\mathrm{d}\mathbf{y}) - \int_\Omega \pi_{2k+2}(\mathrm{d}\mathbf{x}, \cdot)\frac{\mathrm{d}\mu_\star \otimes \mathrm{d}\nu_\star}{\mathrm{d}\mu_{\star,k+1} \otimes \mathrm{d}\nu_{\star,k+1}}, \tag{32}$$

where the last equality follows by the LHS of Eq.(15), the last integral is with respect to $\mathbf{x}$.

Apply Lemma 7 with respect to $\frac{d\mu_\star}{d\mu_{\star,k+1}}(\mathbf{x})$

$$
\begin{aligned}
\int_\Omega \pi_{2k+2}(d\mathbf{x}, \cdot) \frac{d\mu_\star \otimes d\nu_\star}{d\mu_{\star,k+1} \otimes d\nu_{\star,k+1}} &\leq \int_\Omega \big(1 + O(\epsilon)\big)\pi_{2k+2}(d\mathbf{x}, \cdot)\frac{\nu_\star(d\mathbf{y})}{\nu_{\star,k+1}(d\mathbf{y})} \\
&\leq \big(1 + O(\epsilon)\big)\nu_{\star,k+1}(d\mathbf{y})\frac{\nu_\star(d\mathbf{y})}{\nu_{\star,k+1}(d\mathbf{y})} \\
&= \big(1 + O(\epsilon)\big)\nu_\star(d\mathbf{y}),
\end{aligned}
\tag{33}
$$

where the second inequality is derived by the fact that the second marginal of $\pi_{2k+2}$ is $\nu_{\star,k+1}$ in Eq.(14). Similarly, we can show $\int_\Omega \pi_{2k+2}(d\mathbf{x}, \cdot)\frac{d\mu_\star \otimes d\nu_\star}{d\mu_{\star,k+1} \otimes d\nu_{\star,k+1}} \gtrsim (1 - O(\epsilon))\nu_\star(d\mathbf{y})$.

Combining Eq.(32) and (33), we have

$$
|\partial_2 G(\bar\varphi_{k+1}, \bar\psi_{k+1})(\mathbf{y})\nu_\star(d\mathbf{y})| \lesssim \epsilon\nu_\star(d\mathbf{y}).
\tag{34}
$$

We now build the lower bound of the integral as follows

$$
\begin{aligned}
&\int_\Omega \partial_2 G(\bar\varphi_{k+1}, \bar\psi_{k+1})(\mathbf{y})[\bar\psi_{k+1}(\mathbf{y}) - \bar\psi_k(\mathbf{y})]\nu_\star(d\mathbf{y}) \\
&\gtrsim -\epsilon \int_\Omega |\bar\psi_{k+1}(\mathbf{y}) - \bar\psi_k(\mathbf{y})|\nu_\star(d\mathbf{y}) \\
&\gtrsim -\epsilon,
\end{aligned}
\tag{35}
$$

where the first inequality follows by Eq.(32) and the second inequality follows by the boundedness of the potential function in Lemma 4. The above means that $\mathrm{I} \geq \frac{\sigma}{2}\|\bar\psi_k - \bar\psi_{k+1}\|_{L^2(\mu_\star \otimes \nu_\star)} - O(\epsilon)$. For the estimate of II, Lemma 5 yields

$$
\mathrm{II} \geq \int_\Omega \partial_1 G(\bar\varphi_{k+1}, \bar\psi_k)(\mathbf{x})[\bar\varphi_{k+1}(\mathbf{x}) - \bar\varphi_k(\mathbf{x})]\mu_\star(d\mathbf{x}) + \frac{\sigma}{2}\|\bar\varphi_k - \bar\varphi_{k+1}\|_{L^2(\mu_\star)}.
$$

Recall the definition of $\bar\varphi_{k+1}$ in Eq.(28) states that $\int_\Omega e^{\bar\psi_k(\mathbf{y}) - c_\varepsilon(\mathbf{x}, \mathbf{y})}\nu_{\star,k}(d\mathbf{y}) = e^{-\bar\varphi_{k+1}(\mathbf{x}) + \lambda_k}$. Apply Lemma 7 with respect to $\frac{d\nu_\star}{d\nu_{\star,k}}(\mathbf{y})$

$$
\begin{aligned}
\partial_1 G(\bar\varphi_{k+1}, \bar\psi_k)(\mathbf{x}) &= 1 - e^{\bar\varphi_{k+1}(\mathbf{x})}\int_\Omega e^{\bar\psi_k(\mathbf{y}) - c_\varepsilon(\mathbf{x}, \mathbf{y})}\nu_{\star,k}(d\mathbf{y})\frac{\nu_\star(d\mathbf{y})}{\nu_{\star,k}(d\mathbf{y})} \\
&\geq 1 - e^{\bar\varphi_{k+1}(\mathbf{x})}\int_\Omega \big(1 + O(\epsilon)\big)e^{\bar\psi_k(\mathbf{y}) - c_\varepsilon(\mathbf{x}, \mathbf{y})}\nu_{\star,k}(d\mathbf{y}) \\
&\geq 1 - (1 + O(\epsilon))e^{\lambda_k} - O(\epsilon)\underbrace{\int_\Omega \|\mathbf{y}\|_2^2 e^{\bar\varphi_{k+1}(\mathbf{x}) + \bar\psi_k(\mathbf{y}) - c_\varepsilon(\mathbf{x}, \mathbf{y})}\nu_{\star,k}(d\mathbf{y})}_{\text{bounded and non-negative from Lemma 4}} \\
&= 1 - (1 + O(\epsilon))e^{\lambda_k} - O(\epsilon) \\
\partial_1 G(\bar\varphi_{k+1}, \bar\psi_k)(\mathbf{x}) &\leq 1 + (1 + O(\epsilon))e^{\lambda_k} + O(\epsilon),
\end{aligned}
$$

which includes a deterministic scalar (independent of $\mathbf{x}$) and a small perturbation (dependent of $\mathbf{x}$ and $\epsilon$). Denote $R(\mathbf{x}, \mathbf{y}) = \|\mathbf{y}\|_2^2 e^{\bar\varphi_{k+1}(\mathbf{x}) + \bar\psi_k(\mathbf{y}) - c_\varepsilon(\mathbf{x}, \mathbf{y})}$, Combining the centering operation with $\mu_\star(\bar\varphi_{k+1}) = \mu_\star(\bar\varphi_k) = 0$

$$
\begin{aligned}
&\int_\Omega \partial_1 G(\bar\varphi_{k+1}, \bar\psi_k)(\mathbf{x})[\bar\varphi_{k+1}(\mathbf{x}) - \bar\varphi_k(\mathbf{x})]\mu_\star(d\mathbf{x}) \\
&= \text{deterministic scalar} \cdot \underbrace{\int_\Omega [\bar\varphi_{k+1}(\mathbf{x}) - \bar\varphi_k(\mathbf{x})]\mu_\star(d\mathbf{x})}_{:=0 \text{ by the centering operation}} + \epsilon \underbrace{\int_\Omega R(\mathbf{x})[\bar\varphi_{k+1}(\mathbf{x}) - \bar\varphi_k(\mathbf{x})]\mu_\star(d\mathbf{x})}_{\text{integrable by the boundedness of } R, \bar\varphi_{k+1}, \psi_k} \\
&= O(\epsilon).
\end{aligned}
$$

Combining the estimates of I and II completes the proof. $\qquad\square$

### C.1 CONVERGENCE OF DUAL AND POTENTIALS

**Proof of Lemma 2**

**Part I: Convergence of the Dual**

By Lemma 5 with $\alpha = 6\|c\|_\infty$ and the decomposition in Eq.(29), we have

$$
\begin{aligned}
& G(\bar{\varphi}_k, \bar{\psi}_k) - G(\bar{\varphi}_\star, \bar{\psi}_\star) \\
& \geq \int_\Omega \partial_1 G(\bar{\varphi}_k, \bar{\psi}_k)(\mathbf{x})[\bar{\varphi}_k(\mathbf{x}) - \bar{\varphi}_\star(\mathbf{x})]\mu_\star(\mathrm{d}\mathbf{x}) \\
& \quad + \int_\Omega \partial_2 G(\bar{\varphi}_k, \bar{\psi}_k)(\mathbf{y})[\bar{\psi}_k(\mathbf{y}) - \bar{\psi}_\star(\mathbf{y})]\nu_\star(\mathrm{d}\mathbf{y}) \\
& \quad + \frac{\sigma}{2}\left(\|\bar{\varphi}_k - \bar{\varphi}_\star\|_{L^2(\mu_\star)}^2 + \|\bar{\psi}_k - \bar{\psi}_\star\|_{L^2(\nu_\star)}^2\right) \\
& \geq \int_\Omega \partial_1 G(\bar{\varphi}_k, \bar{\psi}_k)(\mathbf{x})[\bar{\varphi}_k(\mathbf{x}) - \bar{\varphi}_\star(\mathbf{x})]\mu_\star(\mathrm{d}\mathbf{x}) + \frac{\sigma}{2}\|\bar{\varphi}_k - \bar{\varphi}_\star\|_{L^2(\mu_\star)}^2 - O(\epsilon),
\end{aligned}
\tag{36}
$$

where $\sigma := e^{-6\|c_\varepsilon\|_\infty}$, and the last inequality follows by Eq.(34) and boundedness of $\bar{\psi}_k$ and $\bar{\psi}_\star$ in Lemma 4. For the first integral, $\int_\Omega \partial_1 G(\bar{\varphi}_{k+1}, \bar{\psi}_k)(\mathbf{x})[\bar{\varphi}_k(\mathbf{x}) - \bar{\varphi}_\star(\mathbf{x})]\mu_\star(\mathrm{d}\mathbf{x}) = O(\epsilon)$ because $\partial_1 G(\bar{\varphi}_{k+1}, \bar{\psi}_k)(\mathbf{x})$ includes a deterministic scalar and a small perturbation with $\mu_\star(\bar{\varphi}_k(\mathbf{x}) = \mu_\star(\bar{\varphi}_\star(\mathbf{x})) = 0$.

Hence

$$
\begin{aligned}
& \int_\Omega \partial_1 G(\bar{\varphi}_k, \bar{\psi}_k)(\mathbf{x})[\bar{\varphi}_k(\mathbf{x}) - \bar{\varphi}_\star(\mathbf{x})]\mu_\star(\mathrm{d}\mathbf{x}) \\
& = \int_\Omega [\partial_1 G(\bar{\varphi}_k, \bar{\psi}_k)(\mathbf{x}) - \partial_1 G(\bar{\varphi}_{k+1}, \bar{\psi}_k)(\mathbf{x})][\bar{\varphi}_k(\mathbf{x}) - \bar{\varphi}_\star(\mathbf{x})]\mu_\star(\mathrm{d}\mathbf{x}) + O(\epsilon) \\
& \geq -\frac{1}{2\sigma}\|\partial_1 G(\bar{\varphi}_k, \bar{\psi}_k) - \partial_1 G(\bar{\varphi}_{k+1}, \bar{\psi}_k)\|_{L^2(\mu_\star)}^2 - \frac{\sigma}{2}\|\bar{\varphi}_k(\mathbf{x}) - \bar{\varphi}_\star(\mathbf{x})\|_{L^2(\mu_\star)}^2 + O(\epsilon),
\end{aligned}
\tag{37}
$$

where the inequality follows from Hölder's inequality and Young's inequality.

Plugging Eq.(37) into Eq.(36), we have

$$
G(\bar{\varphi}_\star, \bar{\psi}_\star) - G(\bar{\varphi}_k, \bar{\psi}_k) \leq \frac{1}{2\sigma}\|\partial_1 G(\bar{\varphi}_k, \bar{\psi}_k) - \partial_1 G(\bar{\varphi}_{k+1}, \bar{\psi}_k)\|_{L^2(\mu_\star)}^2 + O(\epsilon).
\tag{38}
$$

Note that

$$
\begin{aligned}
|\partial_1 G(\bar{\varphi}_k, \bar{\psi}_k)(\mathbf{x}) - \partial_1 G(\bar{\varphi}_{k+1}, \bar{\psi}_k)(\mathbf{x})| & \leq \int_\Omega \left| e^{\bar{\varphi}_{k+1} \oplus \bar{\psi}_k - c_\varepsilon} - e^{\bar{\varphi}_k \oplus \bar{\psi}_k - c_\varepsilon} \right| \nu_\star(\mathrm{d}\mathbf{y}) \\
& \leq e^{6\|c_\varepsilon\|_\infty} \int_\Omega |\bar{\varphi}_{k+1} \oplus \bar{\psi}_k - \bar{\varphi}_k \oplus \bar{\psi}_k| \nu_\star(\mathrm{d}\mathbf{y}) \\
& = \frac{1}{\sigma}|\bar{\varphi}_{k+1}(\mathbf{x}) - \bar{\varphi}_k(\mathbf{x})|,
\end{aligned}
\tag{39}
$$

where the second inequality follows by Lemma 4 and the exponential function follows a Lipschitz continuity such that: $e^a - e^b \leq e^M|b - a|$ for $a, b \leq M$; $\sigma := e^{-6\|c_\varepsilon\|_\infty}$.

First combining Eq.(38) and (39) and then including Lemma 6, we conclude that

$$
\begin{aligned}
G(\bar{\varphi}_\star, \bar{\psi}_\star) - G(\bar{\varphi}_k, \bar{\psi}_k) & \leq \frac{1}{2\sigma^3}\|\bar{\varphi}_{k+1} - \bar{\varphi}_k\|_{L^2(\mu_\star)}^2 + O(\epsilon) \\
& \leq \frac{1}{\sigma^4}\left(G(\bar{\varphi}_{k+1}, \bar{\psi}_{k+1}) - G(\bar{\varphi}_k, \bar{\psi}_k)\right) + \frac{O(\epsilon)}{\sigma^4},
\end{aligned}
$$

where the last inequality follows by $\sigma \leq 1$. Further writing $\Delta_k = G(\bar{\varphi}_\star, \bar{\psi}_\star) - G(\bar{\varphi}_k, \bar{\psi}_k)$, we have

$$
\Delta_k \leq \frac{1}{\sigma^4}\left(\Delta_k - \Delta_{k+1}\right) + \frac{O(\epsilon)}{\sigma^4}.
$$

In other words, we can derive the contraction property as follows

$$\Delta_{k+1} \leq (1 - \sigma^4)\Delta_k + O(\epsilon) \leq \cdots \leq (1 - \sigma^4)^{k+1}\Delta_0 + O(e^{24\|c_\varepsilon\|_\infty}\epsilon).$$

which hereby completes the claim of the theorem for any $k \geq 1$. □

**Part II: Convergence of the Potentials**

For the convergence of the potential function, in spirit to Lemma 6, we obtain

$$G_\star(\bar{\varphi}_\star, \bar{\psi}_\star) - G(\bar{\varphi}_k, \bar{\psi}_k) := \Delta_k \geq \frac{\sigma}{2}\left(\|\bar{\varphi}_\star - \bar{\varphi}_k\|_{L^2(\mu_\star)}^2 + \|\bar{\psi}_\star - \bar{\psi}_k\|_{L^2(\nu_\star)}^2\right) - O(\epsilon).$$

We can upper bound the potential as follows

$$\|\bar{\varphi}_\star - \bar{\varphi}_k\|_{L^2(\mu_\star)}^2 + \|\bar{\psi}_\star - \bar{\psi}_k\|_{L^2(\nu_\star)}^2 \leq \frac{2}{\sigma}\Delta_k + \frac{O(\epsilon)}{\sigma} \leq \frac{2}{\sigma}(1 - \sigma^4)^k\Delta_0 + O(e^{30\|c_\varepsilon\|_\infty}\epsilon).$$

Further applying $(|a| + |b|)^2 \leq 2a^2 + 2b^2$ and $\sqrt{c^2 + d^2} \leq |c| + |d|$, we have

$$\|\bar{\varphi}_\star - \bar{\varphi}_k\|_{L^2(\mu_\star)} + \|\bar{\psi}_\star - \bar{\psi}_k\|_{L^2(\nu_\star)} \leq \sqrt{\frac{4}{\sigma}(1 - \sigma^4)^{k+1}\Delta_0} + O(e^{15\|c_\varepsilon\|_\infty}\epsilon^{1/2}) \tag{40}$$

$$\lesssim e^{3\|c_\varepsilon\|_\infty}\beta_\varepsilon^{\frac{k}{2}} + e^{15\|c_\varepsilon\|_\infty}\epsilon^{1/2},$$

where $\beta_\varepsilon = 1 - \sigma^4 = 1 - e^{-24\|c_\varepsilon\|_\infty}$. □

## C.2 Convergence of the Static Couplings

**Proof of Theorem 2**

Recall from the bounded potential in Lemma 4, we have

$$e^{\bar{\varphi}_\star \oplus \bar{\psi}_\star - c_\varepsilon} - e^{\bar{\varphi}_k \oplus \bar{\psi}_k - c_\varepsilon} \leq e^{6\|c_\varepsilon\|_\infty}\left(|\bar{\varphi}_\star - \varphi_k| + |\bar{\psi}_\star - \bar{\psi}_k|\right). \tag{41}$$

Following Theorem 3 of Deligiannidis et al. (2021), we define a class of 1-Lipschitz functions $\text{Lip}_1 = \{F \big| |F(\mathbf{x}_0, \mathbf{y}_0) - F(\mathbf{x}_1, \mathbf{y}_1)| \leq \|\mathbf{x}_1 - \mathbf{x}_0\|_2 + \|\mathbf{y}_1 - \mathbf{y}_0\|_2\}$. Since the structural property (10) allows to represent $\pi_\star$ using $(\varphi_\star + a) \oplus (\psi_\star - a)$ for any $a$. For any $F \in \text{Lip}_1$, we have

$$\iint_{X \times Y} Fe^{\bar{\varphi}_\star \oplus \bar{\psi}_\star - c_\varepsilon}\mathrm{d}(\mu_\star \otimes \nu_\star) - \iint_{X \times Y} Fe^{\bar{\varphi}_k \oplus \bar{\psi}_k - c_\varepsilon}\mathrm{d}(\mu_{\star,k} \otimes \nu_{\star,k})$$

$$\leq \iint_{X \times Y} Fe^{\bar{\varphi}_\star \oplus \bar{\psi}_\star - c_\varepsilon}\mathrm{d}(\mu_\star \otimes \nu_\star) - \iint_{X \times Y} Fe^{\bar{\varphi}_k \oplus \bar{\psi}_k - c_\varepsilon}\mathrm{d}(\mu_\star \otimes \nu_\star)$$

$$+ \iint_{X \times Y} Fe^{\bar{\varphi}_k \oplus \bar{\psi}_k - c_\varepsilon}\mathrm{d}(\mu_\star \otimes \nu_\star) - \iint_{X \times Y} Fe^{\bar{\varphi}_k \oplus \bar{\psi}_k - c_\varepsilon}\mathrm{d}(\mu_{\star,k} \otimes \nu_{\star,k})$$

$$\leq \iint_{X \times Y} F\underbrace{|e^{\bar{\varphi}_\star \oplus \bar{\psi}_\star - c_\varepsilon} - e^{\bar{\varphi}_k \oplus \bar{\psi}_k - c_\varepsilon}|}_{\text{by Eq.}(41)}\mathrm{d}(\mu_\star \otimes \nu_\star)$$

$$+ \iint_{X \times Y} Fe^{\bar{\varphi}_k \oplus \bar{\psi}_k - c_\varepsilon}\mathrm{d}(\mu_\star \otimes |\nu_\star - \nu_{\star,k}| + |\mu_\star - \mu_{\star,k}| \otimes \nu_{\star,k})$$

$$\lesssim e^{9\|c_\varepsilon\|_\infty}\beta^{k/2} + e^{21\|c_\varepsilon\|_\infty}\epsilon^{1/2},$$

where the last inequality is mainly derived from the first term in the second inequality by combining (40) and (41); the second term in the second inequality can be upper bounded by Lemma 7.

Recall the definition of the duality of the 1-Wasserstein distance, we have

$$\mathbf{W}_1(\pi_k, \pi_\star) = \sup\left\{\iint_{X \times Y} Fe^{\bar{\varphi}_\star \oplus \bar{\psi}_\star - c_\varepsilon}\mathrm{d}(\mu_\star \otimes \nu_\star)\right.$$

$$\left. - \iint_{X \times Y} Fe^{\bar{\varphi}_k \oplus \bar{\psi}_k - c_\varepsilon}\mathrm{d}(\mu_{\star,k} \otimes \nu_{\star,k}) : F \in \text{Lip}_1\right\}$$

$$\leq O(e^{9\|c_\varepsilon\|_\infty}\beta^{k/2} + e^{21\|c_\varepsilon\|_\infty}\epsilon^{1/2}).$$

□

## C.3 Auxiliary Results

**Lemma 7.** *Given probability densities $\rho(\mathbf{x}) = e^{-U(\mathbf{x})}/\mathbb{Z}$ and $\widetilde{\rho}(\mathbf{x}) = e^{-\widetilde{U}(\mathbf{x})}/\widetilde{\mathbb{Z}}$ defined on $\Omega$, where $\overline{\Omega}$ is a bounded domain that contains $\Omega$ and $\widetilde{\Omega}$, $\mathbb{Z}$ and $\widetilde{\mathbb{Z}}$ are the normalizing constants. For small enough $\epsilon \lesssim \frac{1}{(D+1)^2}$, where $D$ is the radius of a centered ball covering $\overline{\Omega}$, we have*

$$1 - O(\epsilon) \leq \frac{\rho(\mathbf{x})}{\widetilde{\rho}(\mathbf{x})} \leq 1 + O(\epsilon), \ 1 - O(\epsilon) \leq \frac{\widetilde{\rho}(\mathbf{x})}{\rho(\mathbf{x})} \leq 1 + O(\epsilon). \tag{42}$$

**Proof**

From the approximation assumption A4: $\|\nabla\widetilde{U}(\mathbf{x}) - \nabla U(\mathbf{x})\|_2 \leq \epsilon(1 + \|\mathbf{x}\|_2)$.

Moreover, $U$ satisfies the smoothness assumption A3. Note that for any $\mathbf{x}, \mathbf{y} \in \Omega$

$$U(\mathbf{x}) - U(\mathbf{y}) = \int_0^1 \frac{\mathrm{d}}{\mathrm{d}t} U(t\mathbf{x} + (1-t)\mathbf{y}) = \int_0^1 \langle \mathbf{x} - \mathbf{y}, \nabla U(t\mathbf{x} + (1-t)\mathbf{y}) \rangle \mathrm{d}t.$$

Moreover, there exist $\mathbf{x}_0$ such that $U(\mathbf{x}_0) = \widetilde{U}(\mathbf{x}_0)$ since $\rho$ and $\widetilde{\rho}$ are probability densities. It follows

$$\begin{aligned}
|\widetilde{U}(\mathbf{x}) - U(\mathbf{x})| &= \left| \int_0^1 \langle \mathbf{x} - \mathbf{x}_0, \nabla\widetilde{U}(\dot{\mathbf{x}}_t) - \nabla U(\dot{\mathbf{x}}_t) \rangle \mathrm{d}t \right| \\
&\leq \int_0^1 \|\mathbf{x} - \mathbf{x}_0\|_2 \cdot \left\| \nabla\widetilde{U}(\dot{\mathbf{x}}_t) - \nabla U(\dot{\mathbf{x}}_t) \right\|_2 \mathrm{d}t \\
&\leq \epsilon(\|\mathbf{x}\|_2 + \|\mathbf{x}_0\|_2)(1 + \|\mathbf{x}\|_2) \lesssim \epsilon(D+1)^2,
\end{aligned}$$

where $\dot{\mathbf{x}}_t = t\mathbf{x} + (1-t)\mathbf{x}_0$ is a line from $\mathbf{x}_0$ to $\mathbf{x}$.

For the normalizing constant, we have

$$|\widetilde{\mathbb{Z}} - \mathbb{Z}| \leq \int_\Omega e^{-U(\mathbf{x})} \big| e^{-\widetilde{U}(\mathbf{x}) + U(\mathbf{x})} - 1 \big| \mathrm{d}\mathbf{x} \lesssim \epsilon \int_\Omega e^{-U(\mathbf{x})} \epsilon(D+1)^2 \mathrm{d}\mathbf{x},$$

where the last inequality follows by Eq.(C.3) and $e^a \leq 1 + 2a$ for $a \in [0, 1]$. We deduce

$$\left| \log \frac{\rho(\mathbf{x})}{\widetilde{\rho}(\mathbf{x})} \right| = \left| \widetilde{U}(\mathbf{x}) - U(\mathbf{x}) + \log \frac{\widetilde{\mathbb{Z}}}{\mathbb{Z}} \right| \leq O(\epsilon).$$

$\square$

Notably, the above lemma also implies that $\mathrm{KL}(\rho\|\widetilde{\rho}) \leq O(\epsilon)$ and $\mathrm{KL}(\widetilde{\rho}\|\rho) \leq O(\epsilon)$.

## C.4 Connections between dual optimization and projections

To see why (12b) holds. We first denote the second marginal of $\mathrm{d}\pi(\varphi_k, \psi_k) := e^{\varphi_k \oplus \psi_k} \mathrm{d}\mathcal{G}$ by $\nu'$ and then proceed to show $\nu' = \nu_\star$ (Nutz, 2022). Recall that $G$ is concave and $\psi_k = \arg\max_{\psi \in L^1(\nu_\star)} G(\varphi_k, \psi)$, it suffices to show that given fixed $\varphi_k \in L^1(\mu_\star)$, $\psi_k \in L^1(\nu_\star)$, a constant $\eta$ and bounded measurable function $\delta_\psi : \mathbb{R}^d \to \mathbb{R}$, the maximality of $G(\varphi_k, \psi_k)$ implies

$$0 = \frac{\mathrm{d}}{\mathrm{d}\eta}\bigg|_{\eta=0} G(\varphi_k, \psi_k + \eta\delta_\psi) = \nu_\star(\delta_\psi) - \iint_{\Omega^2} \delta_\psi e^{\varphi_k \oplus \psi_k} \mathrm{d}\mathcal{G} = \nu_\star(\delta_\psi) - \nu'(\delta_\psi).$$

Hence $\nu' = \nu_\star$. Similarly, we can show (12a).

## C.5 Connections between Static Primal IPF (9) and Static Dual IPF (13)

It suffices to show the equivalence between $\pi_{2k} = \arg\min_{\pi \in \Pi(\cdot, \nu_\star)} \mathrm{KL}(\pi\|\pi_{2k-1})$ and $\psi_k(\mathbf{y}) = -\log \int_\Omega e^{\varphi_k(\mathbf{x}) - c_\varepsilon(\mathbf{x}, \mathbf{y})} \mu_\star(\mathrm{d}\mathbf{x})$.

For any $\pi_{2k} \in \Pi(\cdot, \nu_\star)$, we invoke the disintegration of measures and obtain $\pi_{2k} = \mathrm{K}^? \otimes \nu_\star$. In addition, we have $\pi_{2k-1} = \mathrm{K} \otimes \nu'$. Now we can formulate

$$\mathrm{KL}(\pi\|\pi_{2k-1}) = \mathrm{KL}(\nu_\star\|\nu') + \mathrm{KL}(\mathrm{K}^?\|\mathrm{K}).$$

The conditional probability of $\pi_{2k-1}$ given $\mathbf{y}$ is a normalized probability such that

$$\frac{\frac{\mathrm{d}\pi_{2k-1}}{\mathrm{d}\mu_\star \otimes \nu_\star}}{\int \frac{\mathrm{d}\pi_{2k-1}}{\mathrm{d}\mu_\star \otimes \nu_\star}\mathrm{d}\mu_\star} = \frac{\mathrm{dK}}{\mathrm{d}\mu_\star}(\mathbf{x},\mathbf{y}) = \frac{e^{\varphi_k \oplus \psi_{k-1} - c_\varepsilon}}{\int e^{\varphi_k \oplus \psi_{k-1} - c_\varepsilon}\mathrm{d}\mu_\star} = \frac{e^{\varphi_k - c_\varepsilon}}{\int e^{\varphi_k - c_\varepsilon}\mathrm{d}\mu_\star}.$$

The minimizer is achieved by setting $\mathrm{K}^? = \mathrm{K}$, namely $\pi_{2k} = \mathrm{K} \otimes \nu$. It follows that

$$\frac{\mathrm{d}\pi_{2k}}{\mathrm{d}(\mu_\star \otimes \nu_\star)} = \frac{\mathrm{d}(\mathrm{K} \otimes \nu_\star)}{\mathrm{d}(\mu_\star \otimes \nu_\star)} = \frac{e^{\varphi_k - c_\varepsilon}}{\int e^{\varphi_k - c_\varepsilon}\mathrm{d}\mu_\star} := e^{\varphi_k \oplus \psi_k - c_\varepsilon}.$$

In other words, we have $\psi_k(\mathbf{y}) = -\log \int_\Omega e^{\varphi_k(\mathbf{x}) - c_\varepsilon(\mathbf{x},\mathbf{y})}\mu_\star(\mathrm{d}\mathbf{x})$, which verifies the connections.

## D  EXPERIMENTAL DETAILS

**Reflection Implementations**  Lou & Ermon (2023) already gave a nice tutorial on the implementation of hypercube-related domains with image applications. For extensions to general domains, we provide our solutions in Algorithm 4. The *crucial component* is a *Domain Checker* to verify if a point is inside or outside of a domain, and it appears to be quite computationally expensive. To solve this problem, we propose two solutions :

1) If there exists a computationally efficient conformal map that transforms a manifold into simple shapes, such as a sphere or square, we can apply simple rules to conclude if a proposal is inside a domain.

2) If the first solution is expensive, we can **cache** the domain through a fine-grid mesh $\{\mathrm{X}_{i,j,\cdots}\}_{i,j,\cdots}$. Then, we approximate the condition via

$$\min_{i,j,\cdots} \mathrm{Distance}(\tilde{\mathbf{x}}_{k-1}, \{\mathrm{X}_{i,j,\cdots}\}_{i,j,\cdots}) \leq \text{threshold}.$$

With the parallelism in Torch or JAX, the above calculation can be quite efficient. Nevertheless, a finer grid leads to a higher accuracy but also induces more computations. We can also expect the curse of dimensionality in ultra-high-dimensional problems and simpler domains are more preferred in such cases. Moreover, if $\mathbf{x}_{k+1} \notin \Omega$ in extreme cases, one may consider ad-hoc rules with an error that decreases as we anneal the learning rate. Other elegant solutions include slowing down the process near the boundary or warping the geometry with a Riemannian metric (Fishman et al., 2023).

---

**Algorithm 4** Practical Reflection Operator

---

Simulate a proposal $\tilde{\mathbf{x}}_{k+1}$ via an SDE given $\mathbf{x}_k \in \Omega$.
**if Domain Checker:** $\tilde{\mathbf{x}}_{k+1} \in \Omega$ **then**
    Set $\mathbf{x}_{k+1} = \tilde{\mathbf{x}}_{k+1}$
**else**
    Search (binary) the boundary $\dot{\mathbf{x}}_{k+1} \in \partial\Omega$, where $\dot{\mathbf{x}}_{k+1} = \eta\mathbf{x}_k + (1-\eta)\tilde{\mathbf{x}}_{k+1}$ for $\eta \in (0,1)$.
    Compute $\boldsymbol{\nu} = \tilde{\mathbf{x}}_{k+1} - \dot{\mathbf{x}}_{k+1}$ and the unit normal vector $\mathbf{n}$ associated with $\dot{\mathbf{x}}_{k+1}$.
    Set $\mathbf{x}_{k+1} = \dot{\mathbf{x}}_{k+1} + \boldsymbol{\nu} - 2\langle \boldsymbol{\nu}, \mathbf{n}\rangle\mathbf{n}$.
**end if**

---

### D.1  DOMAINS OF 2D SYNTHETIC DATA

We consider input $t \in [0,1]$ and output $(x,y) \in \mathbb{R}^2$. The normal vector can be derived accordingly.

*Flower* (petals $p = 5$ and move out length $m = 3$)

$$r = \sin(2\pi pt) + m, \quad x = r\cos(2\pi t), \quad y = r\sin(2\pi t).$$

*Heart*

$$x = 16\sin(2\pi t)^3, \quad y = 13\cos(2\pi t) - 5\cos(4\pi t) - 2\cos(6\pi t) - \cos(8\pi t).$$

*Octagon* ($c+1$ edges $(X_i, Y_i)_{i=0}^c$ with $(X_c, Y_c) = (X_0, Y_0)$)

$$r = ct - \lfloor ct \rfloor; \quad x = (1-r)\cdot X_{\lfloor ct \rfloor} + r\cdot X_{\lfloor ct \rfloor+1}; \quad y = (1-r)\cdot Y_{\lfloor ct \rfloor} + r\cdot Y_{\lfloor ct \rfloor+1}.$$

Table 2: Densities using (relected) Langevin diffusion with a practical running time and infinite time.

| SDE | Practical Time | Infinite Time | SDE | Practical Time | Infinite Time |
|-----|----------------|---------------|-----|----------------|---------------|
| VP | Approx. Gaussian | Gaussian | RVP | Approx. Truncated Gaussian | Truncated Gaussian |
| VE | Approx. Gaussian | Uniform | RVE | Approx. Truncated Uniform | Truncated Uniform |

### D.2 HOW TO INITIALIZE: WARM-UP STUDY

Consider a Langevin diffusion (LD) and a reflected Langevin (RLD) with score functions $S_1$ and $S_2$:

$$\text{LD}: \quad \mathrm{d}\mathbf{x}_t = S_1(\mathbf{x}_t)\mathrm{d}t + \mathrm{d}\mathbf{w}_t, \qquad \mathbf{x}_t \in \mathbb{R}^d$$
$$\text{RLD}: \quad \mathrm{d}\mathbf{x}_t = S_2(\mathbf{x}_t)\mathrm{d}t + \mathrm{d}\mathbf{w}_t + \mathbf{n}(\mathbf{x})\mathrm{d}\mathbf{L}_t, \quad \mathbf{x}_t \in \Omega.$$

LD converges to $\mu_1 \propto e^{S_1(\mathbf{x})}$ and RLD converges to $\mu_2 \propto e^{S_2(\mathbf{x})}\mathbf{1}_{\mathbf{x}\in\Omega}$ (Bubeck et al., 2018; Lamperski, 2021) as $t \to \infty$. In other words, inheriting the score function $S_1$ from LD to RLD (by setting $S_1 = S_2$) yields the desired invariant distribution $\mu_1\mathbf{1}_{\mathbf{x}\in\Omega}$.

Denote by RVP (or RVE) the VP-SDE (or VE-SDE) in reflected SB. One can easily show in Table 2 that VP (or RVP) converges approximately to a (or truncated) Gaussian prior within a practical training time, which implies an *unconstrained VP-SDE diffusion model is a good warm-up candidate* for reflected SB. Empirically, we are able to verify this fact through 2D synthetic data.

However, this may not be the case for VE because it converges to a uniform measure in $\mathbb{R}^d$ but only obtains an approximate Gaussian in a short time. By contrast, RVE converges to the invariant uniform distribution much faster because it doesn't need to fully explore $\mathbb{R}^d$. This implies *initializing the score function from VE-based diffusion models for reflected SB may not be a good choice*. Instead, we use RVE-based diffusion models (Lou & Ermon, 2023) as the warm-up.

### D.3 GENERATION OF IMAGE DATA

**Datasets**. Both CIFAR-10 and ImageNet 64×64 are obtained from public resources. All RGB values are between $[0, 1]$. The domain is $\Omega = [0, 1]^d$, where $d = 3 \times 32 \times 32$ for the CIFAR-10 task, $d = 3 \times 64 \times 64$ for the ImageNet task, $d = 1 \times 32 \times 32$ for the MNIST task.

**SDE**. We use reflected VESDE for the reference process due to its simplicity (Lou & Ermon, 2023), and it helps with facilitating the warmup training. The SDE is discretized into 1000 steps. The initial and the terminal scale of the diffusion are $\sigma_{\min} = 0.01$ and $\sigma_{\max} = 5$ respectively. The prior reference is set as the uniform distribution on $\Omega$.

**Training**. The alternate training Algorithm 1 can be accelerated with proper initialization, and the pre-training of the backward score model is critical for successfully training the model. At the warmup phase, the forward score is set as zero and only the backward score model is trained by inheriting the setup in the reflected SGM (Lou & Ermon, 2023). The learning rate is $10^{-5}$. To improve the training efficiency and stabilize the full path-based training target in Algorithm 1, We use Exponential Moving Average (EMA) in the training with the decay rate of 0.99 (Hyvärinen, 2005; Vahdat et al., 2021; Lou & Ermon, 2023).

**Neural networks**. As the high accuracy inference task relies more on the backward score model than the forward score model, the backward process is equipped with more advanced and larger structure. The backward score function uses NCSN++ (Song et al., 2021b) for the CIFAR-10 task and ADM (Dhariwal & Nichol, 2022) for the ImageNet task. The NCSN++ network has 107M parameters. The ADM network has 295M parameters. For MNIST, a smaller U-Net structure with 1.3M parameters (2 attention heads per attention layer, 1 residual block per downsample, 32 base channels) is used for both forward and backward processes. Many previous studies have verified the success of these neural networks in the diffusion based generative tasks (Song et al., 2021b; Lou & Ermon, 2023; Chen et al., 2022b). The forward score function is modeled using a simpler U-Net with 62M parameters.

**Inference**. In both CIFAR-10 and ImageNet 64×64 tasks, images are generated unconditionally, and the quality of the samples is evaluated using Frechet Inception Distance (FID) over 50,000 samples (Heusel et al., 2017; Song et al., 2021b). In MNIST, the the quality of the samples is evaluated using

Negative Log-Likelihood (NLL). Predictor-Corrector using reflected Langevin dynamics is used to further improve the result which does not require any change of the model structure (Song et al., 2021b; Chen et al., 2022b; Lou & Ermon, 2023; Bubeck et al., 2018).

$$\mathbf{x}_t' = \text{reflection}\Big(\mathbf{x}_t + \sigma_t \mathbf{s}(t, \mathbf{x}_t) + \sqrt{2\sigma_t}\varepsilon\Big), \quad \varepsilon \sim N(\mathbf{0}, I)$$

$$\mathbf{s}(t, \mathbf{x}_t) = \frac{1}{g}[\overrightarrow{\mathbf{z}}_t^\theta(t, \mathbf{x}_t) + \overleftarrow{\mathbf{z}}_t^\omega(t, \mathbf{x}_t)], \quad \sigma_t = \frac{2r_{\text{SNR}}^2 g^2 \|\varepsilon\|^2}{\|\mathbf{s}(t, \mathbf{x}_t)\|^2}$$

where $\overrightarrow{\mathbf{z}}_t^\theta, \overleftarrow{\mathbf{z}}_t^\omega$ are the backward and forward score functions as in Algorithm 1.

The likelihood of the diffusion model follows the probabilistic flow neural ODE (Song et al., 2021b; Chen et al., 2022b; Lou & Ermon, 2023)

$$d\mathbf{x}_t = \big[f(\mathbf{x}_t, t) - \frac{1}{2}g(t)^2 \mathbf{s}(t, \mathbf{x}_t)\big]dt := \tilde{f}(\mathbf{x}_t, t)dt$$

$$\log p_0(\mathbf{x}_0) = \log p_T(\mathbf{x}_T) + \int_0^T \nabla \cdot \tilde{f}(\mathbf{x}_t, t)dt$$

Figure 6: Generated samples via reflected SB on MNIST.

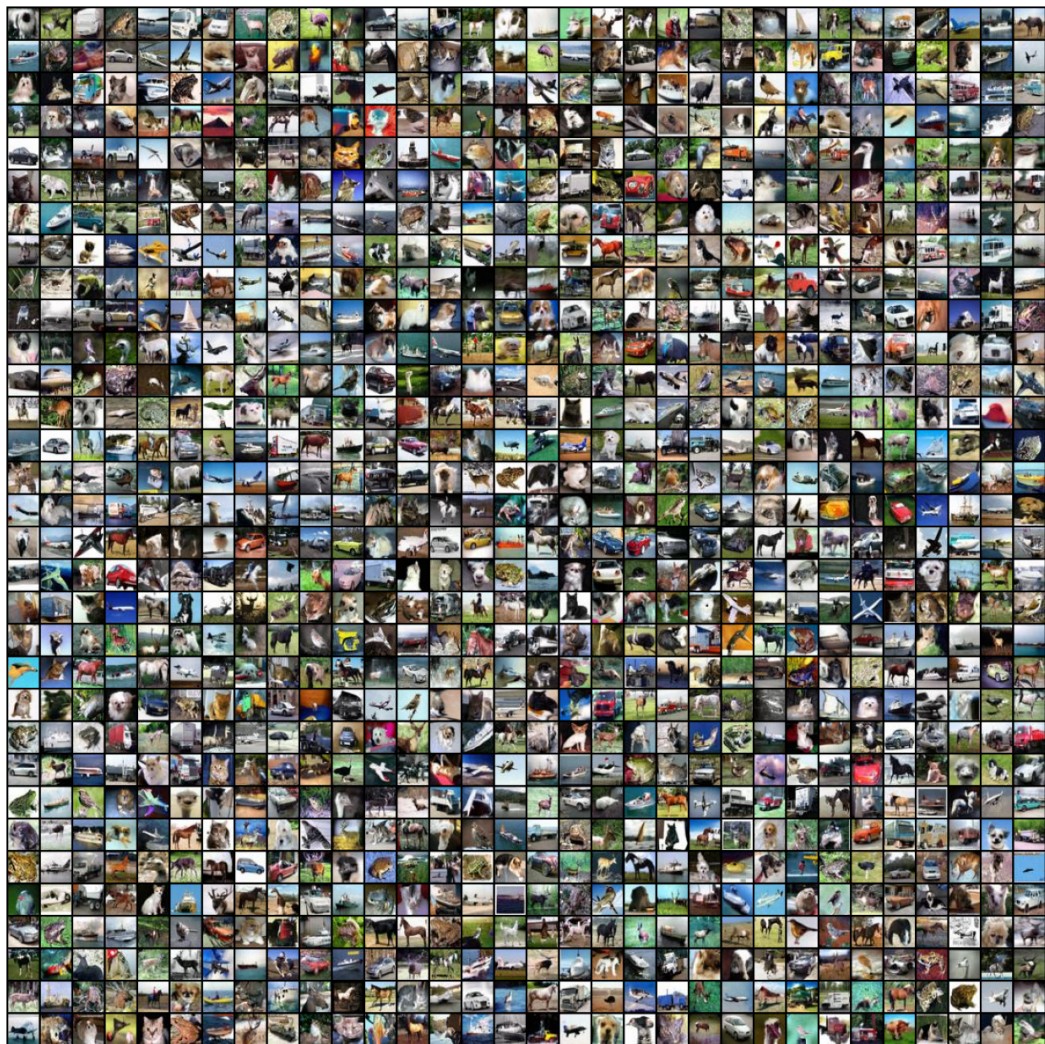

Figure 7: Generated samples via reflected SB on CIFAR-10.

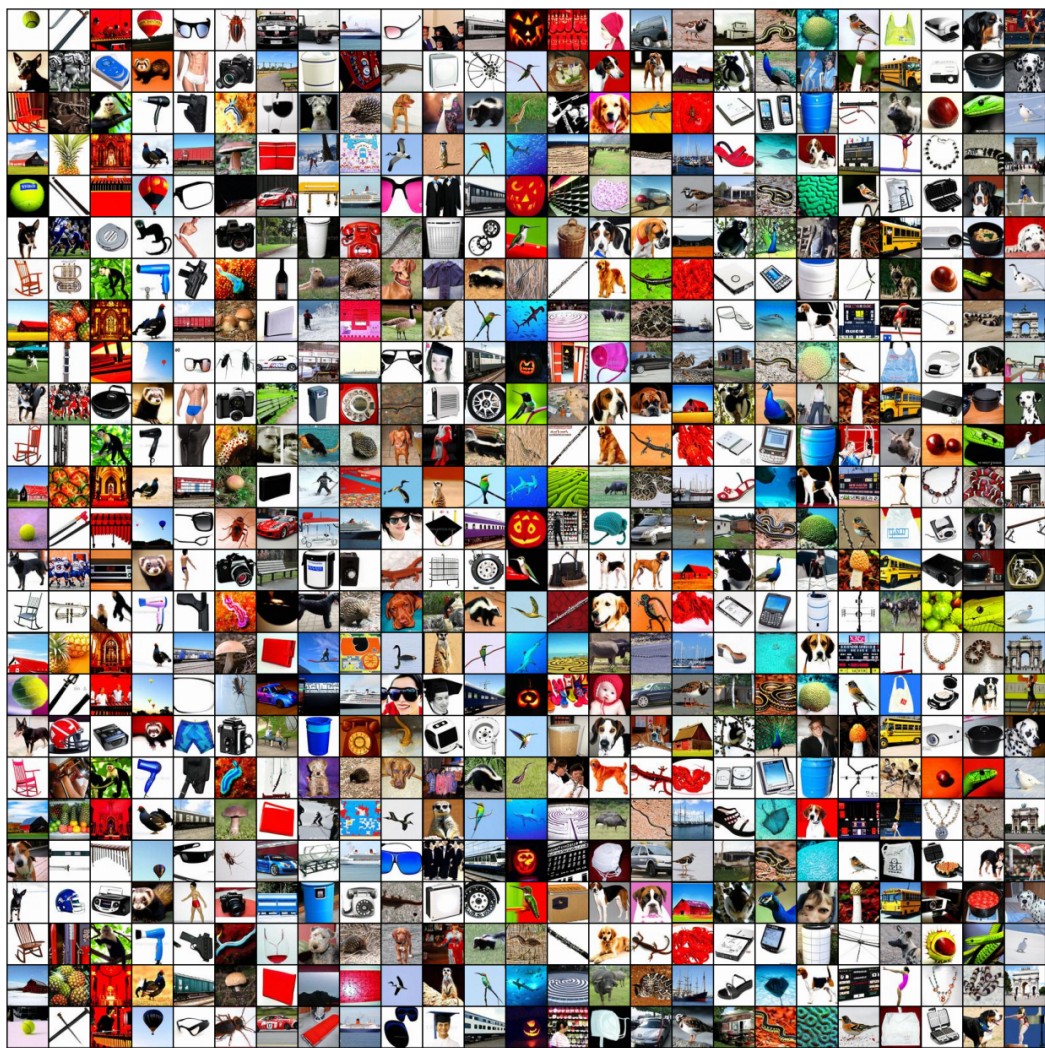

Figure 8: Generated samples via reflected SB on ImageNet-64.

## D.4 GENERATION IN THE SIMPLEX DOMAIN

Alongside the irregular domains illustrated in Figure 2 and the hypercube for image generation, we implement the method on the high-dimensional *projected simplex* using a similar approach as described in Lou & Ermon (2023). A $d$-projected simplex is defined as $\bar{\Delta}_d := \{\boldsymbol{x} \in \mathbb{R}^d : \sum_i \boldsymbol{x}_i \leq 1, \boldsymbol{x}_i \geq 0\}$.

The data is created by collecting the image classification scores of Inception v3 from the last softmax layer with 1008 dimension. All the data fit into the projected simplex $\bar{\Delta}_{1008}$. The Inception model is loaded from a pretrained checkpoint*, and the classification task is performed on the $64 \times 64$ Imagenet validation dataset of 50,000 images. Similar to Lou & Ermon (2023), the diffusion path is constrained within the projected simplex using the stick breaking method as follows:

$$[f(\boldsymbol{x})]_i = \boldsymbol{x}_i \prod_{j=i+1}^{d} (1 - \boldsymbol{x}_j)$$

---

*https://github.com/mseitzer/pytorch-fid/releases/download/fid_weights

The corresponding inverse mapping is,

$$[f^{-1}(\boldsymbol{y})]_i = \frac{\boldsymbol{y}_i}{1 - \sum_{j=i+1}^{d} \boldsymbol{y}_j}$$

In every diffusion step, we first use the reflection operator to limit the data within hypercube, then apply the above stick breaking method to constrain the data within the projected simplex. The neural network of the score function is composed of 6 dense layers with 512 latent nodes.

The results are shown in Figure 8. We compare the generated distribution of the most likely classes. The category index is in the same order of the pre-trained model's output. The last plot in Figure 8 compares the cumulative distribution of the ground truth and generated distribution, providing a cleaner view of the comparison. The curve closely follows the diagonal in the CDF comparison, signifying a strong alignment between the true data distribution and the distribution derived from the generative model.

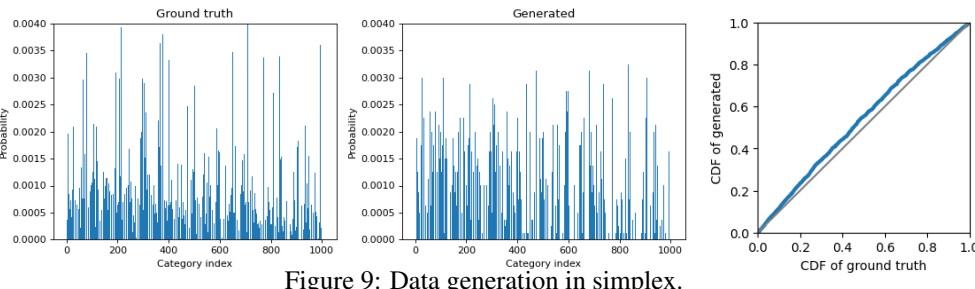

Figure 9: Data generation in simplex.

## D.5 OPTIMAL TRANSPORT HELPS REDUCE NFEs

To demonstrate the effectiveness of OT in reducing the number of function evaluations (NFEs), we study generations based on different NFEs (10, 20, 80) and compare the reflected Schrödinger bridge with reflected diffusion (implicit) (Fishman et al., 2023). We use the same setup (e.g. implicit training loss with the same training budget) to be consistent except that reflected SB uses a well-optimized forward network $\overrightarrow{\mathbf{z}}_t^\theta$ to train the backward network $\overleftarrow{\mathbf{z}}_t^\omega$ while reflected diffusion can be viewed as the first stage of SB training by fixing $\overrightarrow{\mathbf{z}}_t^\theta \equiv \mathbf{0}$.

We observe in Figure 10 that in the regime of NFE=20, both reflected diffusion (implicit) and reflected SB demonstrate remarkably similar generation performance. Despite the inherent compromise in sample quality with smaller NFEs, our investigation revealed that a well-optimized $\overrightarrow{\mathbf{z}}_t^\theta$ significantly contributes to training $\overleftarrow{\mathbf{z}}_t^\omega$ compared to the baseline with $\overrightarrow{\mathbf{z}}_t^\theta \equiv \mathbf{0}$, leading to an improved sample quality even in cases where NFE is set to 10 and 12.

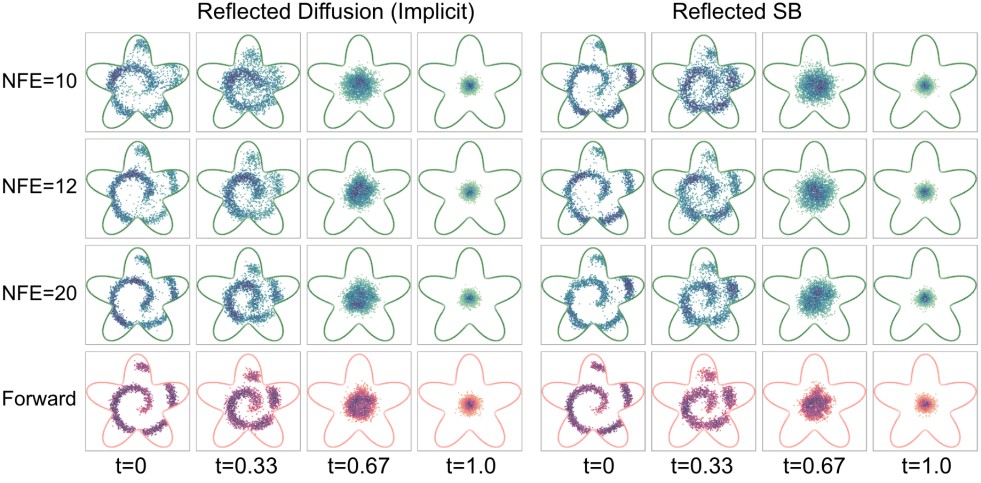

Figure 10: Reflected Schrödinger bridge v.s. reflected diffusion (implicit) based on different NFEs.

