# OpenReview forum: "Reflected Schr\"odinger Bridge for Constrained Generative Modeling"
_ICLR.cc/2024/Conference — Submitted to ICLR 2024_

### Official Review · Reviewer_vVua · 2023-10-31

**Soundness:** 3 good
**Presentation:** 3 good
**Contribution:** 2 fair
**Rating:** 5
**Confidence:** 4

**Summary:**

The authors combine reflected SDEs [1] with the diffusion schrodinger bridge methodology [2] and IPF training scheme. This enables regularized OT on constrained domains. The authors show good empirical performance.

[1] Lou and Ermon Reflected Diffusion Models, 2023 \
[2] Bortoli et al Diffusion Schrodinger Bridge 2023

**Strengths:**

- In general equations and methods appear correct
- This provides a neural entropic OT approach to constrained domains, which is novel as far as I am aware (other diffusion and or simple mapping approaches are not capable of this)
- Generation performance is good for image datasets

**Weaknesses:**

- This is a straight forward connection between reflected SDEs [1] and schrodinger bridge IPF [2,3], without a clear reason or motivation
- Benefits of the method, in particular reduced number of diffusion steps or the benefit of entropic OT on constrained spaces are not really explored, it has not been made clear how this method is beneficial

- Likelihood based training of IPF appears incorrect. The first terms in algorithm 1 involve expected $\log y_T$ and $\log y_0$ starting from points from the alternate marginal distribution $x_0$ and $x_T$ respectively. My understanding form [3] is that these are the result of simulating the diffusion backwards and hence depend on the parameters of the networks. These log terms do not coincide with the log densities of the target measures unless the IPF procedure has converged, hence these one cannot simply ignore them and use this as a loss without taking gradients through the diffusion simulation. I would argue this is not likelihood training and [3]'s argument is incorrect unless there's the assumption that the method has already converged, in which case there is no need for further iterations?

It has been shown by the same authors of [3] that this coincides with regular IPF of [2] and does not require this (incorrect) likelihood derivation.

Minor:

- Section 1: Shi et al 2023 does not use IPF but IMF.
- Vargas et al 2021 does non perform SB on high dimensional examples as claimed, maximum dimension is 4..
- There are many other constrained domain diffusion methods e.g Fishman et al Diffusion Models for Constrained Domains 2023
- The authors mention other related work such as Riemannian SGM Bortoli et al 2022, it should be noted that even more relevant to this work is SB and hence OT has been performed on manifolds in Thornton et al Riemannian Diffusion Schrodinger Bridge, 2022

[1] Lou and Ermon Reflected Diffusion Models, 2023 \
[2] Bortoli et al Diffusion Schrodinger Bridge 2023 \
[3] Chen et al Likelihood Training of Schrödinger Bridge using Forward-Backward SDEs Theory, 2022
[4] Liu et al , Deep Generalized Schrödinger Bridge, 2023

**Edit: correcting typos**

**Questions:**

What are the applications and significance of OT on constrained domains?

Does the proposed method use fewer diffusion steps to obtain similar results to non SB constrained methods?

---

> ### Public Comment · ~Francisco_Vargas1 · 2023-11-16
> **Vargas et al 2021 DID push the frontier towards scaling up IPF to high dimensions**
>
> > Vargas et al 2021 does non perform SB on high dimensional examples as claimed, maximum dimension is 4..
>
> Dear Reviewer,
>
> Thank you for your review of this work I wanted to quickly highlight a clarification here, such as to not demerit/diminish our contribution from 2021.
>
> > Recent advances (De Bortoli et al.,2021; Vargas et al., 2021; Chen et al., 2022b; Shi et al., 2023) have pushed the frontier of IPFs to high-dimensional generative models ..
>
> The algorithm in Vargas et al 2021 et al has allowed to push SBPs to higher dimensions than before in fact if you pay close enough attention you will notice that if replacing the GP with a Neural network the algorithm in our 2021 work is effectively akin to DSB and scales to higher dimensions.
>
> The contribution in 2021 was really to combine ancestral sampling and the chain rule + a regression loss to solve the IPF half bridges, which was entirely novel at the time and allowed for many subsequent works to follow up on this. Both DSB and Vargas et al 2021 brought this contribution concurrently and it is very nice that the authors have decided to acknowledge this.
>
> The authors are not saying that our work specifically explored high dimensions, but that it helped push SBPs to higher dimensions. In fact if you take the time to familiarise yourself with our IPF scheme you will notice just like DSB this scheme consists of a series of non linear regression objectives amenable to both NNs and Kernels, at the time we chose kernels however recent works have shown our exact  IPML scheme to work well in high dimensions (and this is our exact same algorithm):
>
> 1. https://arxiv.org/pdf/2211.01156.pdf (you will see here ML-SB our approach, competitive to DSB)
> 2. https://arxiv.org/pdf/2306.10161.pdf (similar insights here our approach ML-SB/IPML fairs well in high dim comparative to DSB and other works)
>
> So the claim made by the authors that Vargas 2021 aided in scaling SBPs to high dimensions is quite accurate, removing our work from this citation list would be a rather unfair portrayal of our contribution as along with DSB we were the first to propose this family of algorithmic schemes for IPF, small timing note around 2021 when we started this line of work there were no IPF/SBP methods that worked well even in 2 dimensions (we do ablate this in our appendix), for our focus of rare events even 4 dim was very challenging at the time, high dim can also be contextual (varies per task e.g MD, rare events, sampling densities).
>
> > as claimed, maximum dimension is 4..
>
> Also, this claim you have made is **slightly inaccurate** While it's not a big difference but the single-cell dataset which we tackle (a real-world dataset) is 5 dimensional **not 4** :)  (motion capture is indeed 4).
>
> As before thank you for the time in contributing to the reviewing process of ICLR.

---

> ### Comment · Reviewer_vVua · 2023-11-16
>
> Dear Francisco,
>
> Thank you for your interest in this review. I am taken aback by this comment. I hope this is not intentional, but I find your tone and wording quite condescending, and highly unprofessional.
>
> I stand by this remark, regardless of how very minor it is.
>
> If you take the time to read the sentence again - "pushed the frontier of IPFs to high-dimensional generative models" - I do not think this characterises your paper. I do not consider dimension 4 or even 5 to be high dimension. I am not sure anyone would. So in my opinion your work has not "pushed the frontier of IPFs to high-dimensional generative models" as stated in the text, especially when other work at the same time extended IPF to dimensions > 1,000.
>
> I also do not think you showed that your approach can be used for generative modeling.
>
> I agree one could take the regression loss, then adjust the training procedure to be substantially more scalable with a host of training techniques, neural networks, time batching etc. from the diffusion model literature. However to the best of my knowledge this is not done in your work. To cite your work specifically for this would demerit/diminish other contributions from 2021 that actually scale IPF to a dimension in the thousands and not 4 or 5.
>
> I appreciate the contributions of your paper. Indeed it works well for low dimension and non linear reference diffusions. However this minor remark is unrelated to other contributions and does not diminish other contributions. I believe your work should be cited but certainly not for scaling IPF for high dimensional generative modeling.

---

> > ### Author Response · Authors · 2023-11-16
> > **Scalability of GP-based SBs**
> >
> > Dear Francisco  and Reviewer vVua,
> >
> > We would like to express our sincere gratitude for your interest in our work and for providing insightful comments on the scalability of the Gaussian process (GP). Your feedback is invaluable to us.
> >
> > While we maintain our belief that the GP-based SB exhibits scalability to high dimensions, we acknowledge that GP faces challenges in ultra-high-dimensional problems when compared to DNNs. To clarify, we wish to revise our statement: "Recent advances (De Bortoli et al., 2021; Chen et al., 2022b) have pushed the frontier of IPFs to ultra-high-dimensional generative models using DNN frameworks. Additionally, the GP-based SB (Vargas et al., 2021) demonstrates great promise and robustness in high-dimensional generations. Bridge matching methods (Shi et al.,
> > 2023; Peluchetti, 2023) also offers promising alternatives for solving complex dynamic SB problems.
> >
> > Furthermore, we are actively engaged in experiments on the simplex dataset to compare with Lou's 23; we are also trying to demonstrate the advantages in reducing NFEs. We kindly request a few more days to complete these experiments. Your patience is greatly appreciated.

---

> ### Public Comment · ~Francisco_Vargas1 · 2023-11-17
> **thank you for the clarifications.**
>
> Dear Reviewer,
>
> Thank you for your kind response. No intentions to be condescending tone-wise! apologies if it reads as such, written tone is challenging (was aiming for friendly).
>
> The intention was to highlight some aspects that did not seem accurate whether it be how our contribution is portrayed or small details like the dimension we explored on the single-cell dataset (e.g. 5, not 4). So apologies for how the tone was perceived, ideally there should not come across as unprofessional here from my side (I don't think I used any words such as condescending or unprofessional, at least I hope not ! ) was just the hope of an open clarification
>
> To further reassure there is no intention to devalue your review which I actually found remarkably helpful when reading this work and as before thank you for your review.
>
> > I  do not think this characterizes your paper. I do not consider dimension 4 or even 5 to be high dimension. I am not sure anyone would.
>
> This can sound slightly dismissive in particular "I am not sure anyone would" but I am sure the reviewer does not intend it as such. Difficulty and dimension can depend on context, if you consider rare events problems in molecular dynamics, already going above 3 dimensions is quite high and difficult to solve (e..g trying to sample a transition path between two metastable configurations).  So again it really depends on context and task. I concede the point that it is definitely not high for standard gen modeling.
>
> > I also do not think you showed that your approach can be used for generative modeling.
>
> We highlighted its potential and did some very minor toy experiments (low dim gen modeling) within this context but agreed we definitely did not push or focus fully on this front.
>
> >  believe your work should be cited but certainly not for scaling IPF for high dimensional generative modeling.
>
> I still stand with my initial high-level remark, whilst also agreeing with some of your very careful sub-remarks:
>
> 1. 100% agreed that our focus was not generative modeling, so I do see where you are coming from overal, and definitely worth removing it from that cite list on that point of focus.
> 2. Our algorithm allows IPF to be scaled to higher dimensions and again was one of the earliest to do so. My point here is that we do place in the list of works/algorithms that pushed IPF towards high dimensional settings.
>
> If a work provides a general algorithm that allows for a scalable high-dimension implementation (which has been verified ) then it definitely aided in pushing the frontier and as highlighted before our proposed algorithm has since then (e.g. https://arxiv.org/pdf/2211.01156.pdf ) been verified to scale well to higher dimensions.  Furthermore whilst our focus was on GPs at the time we emphasized in the original our proposed scheme could also be trained with NNs rather than GPs, the point here being our high-level algorithm can handle high dimensions well as empirically explored in https://arxiv.org/pdf/2211.01156.pdf (without major tricks or adjustments, they did in fact use our codebase).
>
> >  .. certainly not for scaling IPF for high dimensional generative modeling.
>
> Definitely agree with you and I see the importance of citing accurately such as not to shadow other contributions.  But the wording was a bit different here in particular the notion of pushing the frontiers towards high dim IPF, is something that I do believe our work contributed towards by laying down a scalable algorithm.
>
> All this said, again I have to thank the reviewer for taking the time to review this work and also for calmly explaining their reasoning which is incredibly kind of them, especially during this hectic period.
>
> Also want to highlight that this is very exciting work so I really do not want to distract any more from the review process here.
>
> Enormous apologies to the authors for derailing the conversation, and thank you for this very exciting work.

---

> ### Author Response · Authors · 2023-11-18
> **Connections, losses, and experiments**
>
> We appreciate the reviewer for the insightful suggestions.
>
> **Straightforward connections between reflected SDEs and Schrodinger bridge (SB)**
>
> While the SB algorithms (Valentin'21, Chen'22) have made significant strides in the study of dynamic optimal transport and generative modeling, our understanding of constrained generative modeling with non-linear transport and theoretical optimal transport guarantees remains incomplete, especially in light of real-world data often exhibiting bounded support. To address this challenge, we derive reflected forward-backward stochastic differential equations (reflected FB-SDEs) with Neumann and Robin boundary conditions, yielding a specific loss function for constrained generative modeling on arbitrary smooth domains. Furthermore, we establish connections between reflected FB-SDEs and Entropic Optimal Transport (EOT) on bounded domains. Notably, **our work opens avenues for analyzing the linear convergence of dynamic couplings on bounded domains, offering valuable insights into high-dimensional generative models.**
>
> **Log terms do not coincide with log densities unless the IPF procedure has converged and Why Chen'22's alternative training has no $E[\log \overleftarrow y_T]$ and $E[\log \overrightarrow y_0]$, while ours does.**
>
> Although the likelihood in Proposition 1 is theoretically correct, we recognize the challenges in estimating $E[\log \overleftarrow y_T]$ and $E[\log \overrightarrow y_0]$ in Alg.1, particularly at non-differentiable terminal time points $t=0$ and $T$. A more pragmatic alternative involves omitting these terms following Chen'22, albeit at the cost of a slight increase in the variational gap. We have made the corresponding revision in Alg.1. We express our gratitude to reviewer vVua again for capturing the important details and improving the algorithmic demonstration.
>
> **Empirical justification of Fewer NFEs**
>
>
> We have included section D.5 in the appendix to study how optimal transport helps reduce NFEs in a simulation example. We note that optimizing the forward network plays a crucial role in training the backward network, especially when contrasted with the baseline where the forward network weights are set to 0. The well-optimized forward network leads to an improved sample quality even in cases where NFE is set to 10 and 12. We also wish to produce more examples in the final version.
>
>
>
> **Comments on the References**
>
> We value your insightful suggestions regarding the references. In response, we have reintroduced the bridge matching method in the introduction section and incorporated the more pertinent reference [1] into our related works.
>
> [1] Riemannian Diffusion Schrödinger Bridge.
>
> We sincerely hope we have addressed your concerns and kindly request the reviewer to generously reconsider the work.

---

### Official Review · Reviewer_bpMp · 2023-10-31

**Soundness:** 3 good
**Presentation:** 2 fair
**Contribution:** 2 fair
**Rating:** 3
**Confidence:** 3

**Summary:**

In this paper, the authors propose to extend the Schrodinger Bridge methodology to state space which are defined by constraints. They follow closely the methodology based on Forward Backward Stochastic Differential Equations (FBSDEs) introduced in [1]. The main contribution of the paper is to extend this set up to the case where the FBSDEs are replaced with a reflected FBSDE. This is in accordance with the recent works of [2,3] who study reflected versions of diffusion models. Another contribution of the paper is to study the convergence of Iterative Proportional Fitting (IPF) approaches. In that setting, the authors follow [4]. The authors also illustrate the efficiency of their method with

[1] Chen et al. (2021) -- Likelihood Training of Schrödinger Bridge using Forward-Backward SDEs Theory

[2] Lou and Ermon (2023) -- Reflected Diffusion Models

[3] Fishman et al. (2023) -- Diffusion Models for Constrained Domains

[4] Chen et al. (2023) -- Provably Convergent Schrödinger Bridge with Applications to Probabilistic Time Series Imputation

**Strengths:**

* The extension of the Schrodinger Bridge framework to the reflected setting is new to my knowledge.

* The convergence guarantees presented in the paper are also interesting (although see my comments in the "Weaknesses" section for some concerns I have regarding their novelty and soundness).

* I appreciate the fact that the authors produce extensive experiments and compare themselves to [1].

[1] Lou and Ermon (2023) -- Reflected Diffusion Models

**Weaknesses:**

* My main concern is with the motivation of the paper. While it is certainly possible (as shown by the authors) to extend the Schrodinger Bridge framework to the reflected setting it is not clear why one would like to do this. I think that a contribution to a top tier venue such as ICLR requires more motivation (either theoretical, methodological or experimental), note that I am not talking about the novelty here (there is no doubt that the presented work is novel). It seems that the authors want to put forward that Schrodinger Bridge has the ability to reduce the length of the integration time in generative modeling (a reason first put forward in the work of [1]) "Notably, the forward process (1a) requires a long time T to approach the prior distribution, which inevitably leads to a slow inference". However there already exist many works focused on improving the speed of diffusion models (distillation, better samplers, etc.) which have proven to be more efficient than Schrodinger bridges.

* The authors claim that Proposition 1 is a valid ELBO, however this result is only true at equilibrium. It is no longer true when the vector fields are parametric functions.

* I am not very convinced by the significance of the empirical results. As I emphasized earlier, I appreciate the fact that the authors provide an extensive image investigation, however the results are not very convincing. The quality seems to be worse than [2]. If the authors claim that the reflected SB method improves the speed of the generation then it would have been useful to compare NFE.

* It is not clear at all for me what are the differences between the theoretical section and the work of [3]. The work of [3] is cited but not comparison or discussion is provided.

* The assumption A4 is very strong. It is claimed that it is similar to the one of [4] but this is not true. In fact, the assumption of [4] only holds for one iteration of the IPF. The stability of the approximation across IPF iterations is not clear at all and should be discussed.

Minor comments:

* In the introduction [5] should be cited when citing [6] as the works are concurrent.

* In (3) missing index $t$ in $x_t$

* When deriving the reflected FBSDE framework I would have appreciated more emphasis on the introduced quantities.

* Could the authors be more precise on the algorithmic differences between the introduced algorithm and the one of [7]?

* It would be very interesting to understand if the current setting can accomodate for an understanding of the dynamic thresholding procedure used in [8]

[1] De Bortoli et al. (2021) -- Diffusion Schrödinger Bridge with Applications to Score-Based Generative Modeling

[2] Lou and Ermon (2023) -- Reflected Diffusion Models

[3] Chen et al. (2023) -- Provably Convergent Schrödinger Bridge with Applications to Probabilistic Time Series Imputation

[4] De Bortoli (2022) -- Convergence of diffusion models under the manifold hypothesis

[5] Peluchetti (2023) -- Diffusion Bridge Mixture Transports, Schrödinger Bridge Problems and Generative Modeling

[6] Shi et al. (2023) -- Diffusion Schrödinger Bridge Matching

[7] Chen et al. (2021) -- Likelihood Training of Schrödinger Bridge using Forward-Backward SDEs Theory

[8] Saharia et al. (2022) -- Photorealistic Text-to-Image Diffusion Models with Deep Language Understanding

**Questions:**

Please answer to the main comments in the "Weaknesses" section. The authors should focus on the motivation of their method. As of now, it is not clear what are the key contributions of the paper.

---

> ### Author Response · Authors · 2023-11-18
> **Motivation, ELBO, and Assumptions A4.**
>
> We appreciate the reviewer for the valuable comments.
>
> **Major Novelty and Motivation of the paper**
>
> Our major novelty comes from the constrained generation on **arbitrary smooth** bounded domains with **theoretical support**. To achieve this goal, we developed the reflected Schrodinger bridge algorithm (rSB), which provides the first work of generative models that extends to arbitrary smooth geometries with optimal transport guarantees. Theoretically, we establish the elegant connections between rSB and the linear convergence of optimal transport on bounded domains. This not only enhances our understanding of the algorithm but also lays the foundation for exploring the theoretical properties of dynamic Schrödinger bridges—a domain that remains less explored. This novel linkage contributes significantly to the field, providing a deeper comprehension of the algorithm and paving the way for the study of dynamic Schrödinger bridges.
>
> **Proposition 1 is not a valid ELBO when vector fields are parametric functions**
>
> We respectfully **disagree** with this argument. Firstly, we did not claim that Proposition 1 constitutes an Evidence Lower Bound (ELBO), as it is not parameterized by $\overrightarrow z_t^{\theta}$ and $\overleftarrow z_t^{\omega}$; rather, it serves as the precise likelihood /loss function. Secondly, the statement that the parametrized loss bears resemblance to the ELBO is initially posited in Theorems 1 and 2 in [1], and this statement holds true in our context as well.
>
> [1] Song, Maximum Likelihood Training of Score-Based Diffusion Models. 2021
>
> **Difference between the theoretical section and the work of [3]**
>
> We want to emphasize that the convergence of optimal transport in **general domains and bounded domains are fundamentally different**. For instance, [1] delved into linear convergence on bounded domains, whereas [2] investigated sublinear convergence in general domains.
>
> Concerning the convergence of optimal transport with practically approximated marginals, [3] explored the **primal** Iterative Proportional Fitting (IPF) in a **general** space with a **sublinear** rate. In contrast, our work examines the convergence of the **dual** IPF in a **bounded** space with a **linear** rate. It is noteworthy that our study on the reflected Schrödinger bridge also reveals the linear convergence of optimal transport on bounded domains, offering a more practical and **tangible impact and guiding the hyperparameter tuning of $\varepsilon$**.
>
> [1] Guillaume Carlier (2022). On the Linear Convergence of the Multi-Marginal Sinkhorn Algorithm.
>
> [2] Ghosal and Nutz (2022) On the Convergence Rate of Sinkhorn’s Algorithm.
>
> [3] Chen (2023) Provably Convergent Schrodinger Bridge with Applications to Probabilistic Time Series Imputation.
>
>
> **Assumption A4 is very strong**
>
> We respectfully **disagree** with this argument. A4 is a mild assumption, akin to A3 in [1], quantifying the distance between distributions. However, these assumptions serve distinct purposes. [1] establishes convergence in score matching under the manifold hypothesis with A3 quantifying marginal distribution error. Our theory focuses on demonstrating the stability of Iterative Proportional Fitting (IPF) for Schrödinger bridge regarding marginal distributions, with A4 modeling perturbations in the marginals.
>
>
> As shown in Figure 3, standard IPF iterates require exact alternating projections such that $\mu_{2k+1}=\mu_{\star}$ and $\nu_{2k}=\nu_{\star}$ (see Eq.(6.5) in page 55 of [2]) in **every iteration instead of only the first iteration**. Computing the exact projections and obtaining the exact marginal $\mu_{\star}$ (or $\nu_{\star}$) at $2k+1$ (or $2k$) iterations for any $k=1,2,3, 4$ can be expensive in practice. To tackle this issue, we relax the requirement to approximated projections and marginals in Eq(14) such that $\mu_{2k+1}=\mu_{\star, k+1}$ and $\nu_{2k}=\nu_{\star, k}$, where the scale of perturbations is quantified by A4.
>
> Consistent observations were made in [3], which employed a geometric approach rather than a probabilistic one. Additionally, our approach allows for more flexible perturbations across different IPF iterations.
>
> **We consider our Assumption A4, which quantifies the perturbation, to be standard. Any identification of major errors in our proof would be greatly appreciated.**
>
> [1] De Bortoli (2022) -- Convergence of diffusion models under the manifold hypothesis.
>
> [2] Marcel Nutz (2022). Introduction to Entropic Optimal Transport.
>
> [3] Deligiannidis (2021), etc. Quantitative Uniform Stability of the Iterative Proportional Fitting Procedure. 2108.08129v2

---

> ### Author Response · Authors · 2023-11-18
> **Fewer NFEs, ours v.s. Chen22, and others**
>
> **Empirical justification of Fewer NFEs**
>
> Experimentally, we acknowledge that the class of Schrodinger bridge algorithms is expensive and may not provide compelling large-scale empirical results compared to techniques such as distillation. We have included section D.5 in the appendix to study the reduced NFEs in a simulated example, which shows that a well-optimized forward network significantly contributes to training the backward network compared to the baseline with the forward network weights being 0, which leads to an improved sample quality even in cases where NFE is set to 10 and 12. We wish to produce more examples in the final version.
>
> **Difference between our algorithm and the algorithm in Chen22**
>
> Our algorithm represents a natural yet crucial extension of Chen's algorithm for constrained generation on smooth domains (Chen22). The primary distinction lies in the incorporation of local time in the simulations and the Neumann and Robin boundary conditions in the derivation of reflected FB-SDEs. Additionally, we emphasize that working with bounded domains offers a **more favorable linear convergence rate** for guiding the hyperparameter tuning of $\varepsilon$ in practice. Such a property **does not hold** for general domains in Chen22.
>
>
> Minor
>
> **missing one literature on bridge matching**
>
> Thank you for bringing attention to the noteworthy work on bridge matching studies. We have incorporated this relevant contribution into the revised version.
>
> **If the algorithm accommodates dynamic thresholding**
>
> Yes. Similar to the study of Corollary A.11 in [1], dynamic thresholding approximates the projection operator on hypercubes given a fine enough discretization and inner pointing drifts.
>
> [1] Lou (2023) Reflected Diffusion Models.
>
> We sincerely hope we have addressed your detailed concerns and kindly request the reviewer to generously re-evaluate the work.

---

> > ### Author Response · Authors · 2023-11-22
> > **Discussions**
> >
> > Dear Reviewer bpMp,
> >
> > In light of the limited available time, we would greatly appreciate it if you could engage in a discussion with us. We believe that an open dialogue will facilitate a clearer understanding of our responses and allow us to make any necessary revisions promptly.
> >
> > Thanks.

---

### Official Review · Reviewer_qZiS · 2023-10-31

**Soundness:** 4 excellent
**Presentation:** 3 good
**Contribution:** 3 good
**Rating:** 5
**Confidence:** 4

**Summary:**

This paper generalizes shrodinger bridge models to constrained domains. The construction uses reflected brownian motion to construct the diffusion processes and an IPF procedure to fit the machine learning component. Compared with the previous work on score based reflected diffusion models, this method has the added benefit of being more generalizable across geometries and base distributions.

**Strengths:**

* The proposed framework is technically sound and helps overcome fundamental issues with working with the more complex geometry of the base space.
*  The technical constructions are constructed very cleanly with a full explication of the strengths and limitations of the underlying model.
* The technical analysis showcase theoretical speedups for convergence, which can be important on more complicated geometric base spaces.

**Weaknesses:**

* I wouldn't say that [1] doesn't necessarily propose approaches to deal with general geometries. In particular, they propose a diffeomorphic mapping scheme that should help generalization to other spaces although there are of course analytic blowups at edges/corners.
* There could be more experimental results. In particular, the current results mostly show that the method works and can scale to high dimensions. Importantly, a more involved example (e.g. high dimensional simplex) would be needed to show that the method retains its niceties, as compared with [1].

[1] Reflected Diffusion Models Lou and Ermon

**Questions:**

N/A

---

> ### Author Response · Authors · 2023-11-18
> **Refined description of reflected diffusion and more experiments.**
>
> We appreciate the reviewer for the insightful suggestions.
>
> **Reflected diffusion model may also tackle general domains.**
>
> We have refined our description of reflected diffusion models in the abstract and conclusion sections. For example, instead of saying "reflected diffusion models lack the flexibility to adapt to diverse domains", we wrote "reflected diffusion models may not easily adapt to diverse domains without the derivation of proper diffeomorphic mappings" to emphasize the flexible constrained generation ability when diffeomorphic mappings are available.
>
> **More experimental results, including a high dimensional simplex result.**
>
> We have included section D.4 to study the generation in the simplex domain, which obtained a similar promising performance compared with Lou'23; we also observe that a well-optimized forward network helps in reducing the number of function evaluations in a simulation example and results in a better sample quality in section D.5
>
>
> We sincerely hope we have addressed your concerns and kindly request the reviewer to re-evaluate the work.

---

> > ### Comment · Reviewer_qZiS · 2023-12-01
> > **Thank you for your response.**
> >
> > I have read the author response as well as the response to other reviewers.
> >
> > I am a little disappointed that, for the simplex example, it seems that the method is still reliant on the stick-breaking diffeomorphic mapping procedure. I understand that the authors are unable to respond since it's a bit later, but I will take their comments into accounting when discussing with the other reviewers.

---

> ### Public Comment · ~Unknown_Authors1 · 2023-12-05
> **The reflection operator is more stable than the stick-breaking method**
>
> By leveraging the public code, we can effortlessly replicate the simplex experiments using the reflection operator without resorting to diffeomorphic mappings. The obtained results are showcased in the following link: https://pasteboard.co/p3nLOh4AJx95.png . Notably, our findings reveal that the reflection operator outperforms diffeomorphic mappings in generating high-dimensional projected simplices. This is particularly evident when compared to the stick-breaking method, which, when utilizing diffeomorphic mappings, exhibits noticeable distribution bias attributed to analytic blowups at edges and corners.

---

### Author Response · Authors · 2023-11-18
**Official Comment by Authors**

We sincerely value the time and effort dedicated to reviewing our paper. In response to the insightful feedback provided by the reviewers, we have revised the manuscript, highlighting the updated sections in a distinctive orange color. We kindly request you to review the revised version of our paper at your earliest convenience. Your continued feedback is greatly appreciated. Thank you.

We aim to address the common questions in our comprehensive response.

**Empirical justification of Fewer NFEs**

We have included section D.5 in the appendix to study how optimal transport helps reduce NFEs in a simulation example. We note that optimizing the forward network plays a crucial role in training the backward network **with smaller NFEs**, especially when contrasted with the baseline where the forward network weights are set to 0. The well-optimized forward network leads to an improved sample quality even in cases where NFE is set to 10 and 12. We also wish to produce more examples in the final version.

**Proposition 1 and the training loss in Algorithm 1**


Although the likelihood in Proposition 1 is theoretically correct, we recognize the challenges in estimating $E[\log \overleftarrow y_T]$ and $E[\log \overrightarrow y_0]$ in Alg.1, particularly at non-differentiable terminal time points $t=0$ and $T$. A more pragmatic alternative involves omitting these terms, albeit at the cost of a slight increase in the variational gap. Such an approach aligns with the practice advocated by Chen'22. We have made the corresponding revision in Alg.1 in the revised version. We express our gratitude to reviewer vVua for providing insightful suggestions that helped us capture the details.

**(Major) Errata:**

1. Refine the description of reflected diffusion models on general domains and rephrase the literature on the scalability of IPFs; Add a few related works.

2. Add a common treatment for $E[\log \overleftarrow y_T]$ and $E[\log \overrightarrow y_0]$ in Algorithm 1.

3. Add section D.4 to study generation in the simplex domain.

4. Add section D.5 to study how optimal transport helps reduce NFEs.

---

### Meta-Review · Area_Chair_oX15 · 2023-12-07

**Metareview:**

The paper generalizes schrodinger bridge models to constrained domains by combining reflected SDEs, the
 diffusion schrodinger bridge methodology and IPF training scheme.

Most reviewers have found the extension interesting and relevant with nice theoretical contributions.
However, they are overall lukewarm about the paper due to several minor issues. they believe that the
paper still deserves some improvements before acceptance. We thus encourage authors to take into
account the reviewers comment before resubmission to major venues.

**Justification For Why Not Higher Score:**

there is still several issues (theoretical, experimental)  on the paper that has not been addressed by the rebuttal

**Justification For Why Not Lower Score:**

n/a

---

### Decision · Program_Chairs · 2024-01-16

Reject